# An antagonistic epigenetic mechanism regulating gene expression in pollen revealed through single-nucleus multiomics

Colette L. Picard [1,6], Lucia Ichino [1,2,6], Tyler J. Buckley [1,2], Brandon A. Boone [1,2,5], Jaewon Yun[1], Kevin D. Abuhanna[3], Yi Zhang[3], Noah J. Behrendt[1], Su May Lei Soe [1], Chongyuan Luo [3] & Steven E. Jacobsen [1,2,4] ✉

Arabidopsis MBD5, MBD6, and MBD7 are CG-specific methyl-readers with opposite functions: MBD5 and MBD6 (MBD5/6) repress methylated loci in pollen vegetative nuclei (VN), while MBD7 prevents transgene silencing, possibly by promoting DNA demethylation. Here we show that loss of *MBD7* rescues transcriptional defects at a large subset of MBD5/6-bound loci. Using simultaneous profiling of DNA methylation and transcription in single pollen nuclei, we found that MBD5/6-bound loci that are actively demethylated in immature VN lose additional methylation in *mbd5/6*, prior to transcriptional derepression. A subset of these loci is also bound by MBD7, correlating with demethylation and transcriptional derepression in *mbd5/6* that are both reversed by loss of *MBD7*. Conversely, ectopically recruiting the MBD7 complex to MBD5/6 targets causes partial demethylation and upregulation. We propose that MBD5/6 maintain silencing in VN in part by preventing the MBD7 complex from enhancing the active demethylation that occurs during VN maturation.

DNA methylation is an epigenetic modification associated with transcriptional silencing in eukaryotic organisms[1–3]. DNA methylation can repress transcription by interfering with transcription factor binding or by recruiting specific repressors to chromatin[1,3]. In *Arabidopsis thaliana*, two CG-specific DNA methylation readers, MBD5 and MBD6, act redundantly to maintain silencing of a group of transposable elements and promoter-methylated genes[4]. MBD5 and MBD6 interact with α-crystallin domain (ACD) containing protein ACD15 via their StkyC domains[5]. ACD15 in turn dimerizes with ACD21, which recruits the J-domain protein SILENZIO (SLN)[5]. All members of the protein complex are required for silencing[5]. Although MBD5, MBD6, and their interactors are expressed throughout the plant, transcriptional derepression in *mbd5 mbd6* double mutants (*mbd5/6*) is mostly restricted

to the vegetative nucleus (VN) of pollen[6]. The VN has a distinctive epigenetic state, characterized by loss of CG methylation due to active demethylation primarily by DEMETER (DME) and to a lesser extent ROS1, loss of repressive histone modification H3K9me2, and chromatin decondensation[7–14]. The VN specificity of *mbd5/6* derepression suggests that MBD5/6 are uniquely important in this altered environment, although the mechanism remains unclear[6].

Unlike MBD5/6, MBD7 prevents silencing of transgenes and a limited number of endogenous genes[15–18]. MBD7 also recruits a pair of ACD proteins via its StkyC domain: INCREASED DNA METHYLATION 3 (IDM3) (or LIL) and IDM2, which in turn interact with the histone acetyltransferase IDM1[15–18]. In seedlings, loss of each component of the MBD7 complex causes the same mild DNA hypermethylation and

[1]Department of Molecular, Cell and Developmental Biology, University of California Los Angeles, Los Angeles, CA, USA. [2]Molecular Biology Institute, University of California Los Angeles, Los Angeles, CA, USA. [3]Department of Human Genetics, University of California Los Angeles, Los Angeles, CA, USA. [4]Howard Hughes Medical Institute (HHMI), UCLA, Los Angeles, CA, USA. [5]Present address: Department of Chemistry and Physics, Western Carolina University, Cullowhee, NC, USA. [6]These authors contributed equally: Colette L. Picard, Lucia Ichino. ✉e-mail: jacobsen@ucla.edu

transcriptional downregulation phenotype[15–23]. The MBD7 complex was proposed to help promote the activity of DNA demethylases at CG-dense methylated DNA, possibly via the histone acetyltransferase activity of IDM1, leading to loss of DNA methylation and increased expression[18,20–23]. Alternatively, since DNA methylation changes in MBD7 complex mutants are modest, it has also been proposed that MBD7 may function purely downstream of DNA methylation[15]. It is also possible that the function of the MBD7 complex is more pronounced in specific tissues.

MBD5/6 and MBD7 all bind genomic regions characterized by a high density of methylated CG dinucleotides[4,18], raising the question of how these two complexes functionally interact. Here we report that loss of *MBD7* rescues transcriptional derepression in *mbd5/6* VN at >50% of loci. Using simultaneous profiling of DNA methylation and RNA expression in single pollen nuclei, we show that CG methylation is decreased in *mbd5/6* pollen specifically at sites demethylated by DME/ROS1 in the VN, and this occurs prior to transcriptional derepression. Both the methylation loss and the transcriptional derepression are rescued in *mbd5 mbd6 mbd7 (mbd5/6/7)* at a subset of MBD5/6 targets that are more densely methylated and particularly enriched in MBD7 ChIP-seq signal. Thus, MBD5 and MBD6 prevent excessive demethylation in the VN at shared DME/ROS1 and MBD7 targets by antagonizing MBD7. MBD7 binding to chromatin is not directly affected by MBD5/6. We therefore propose that the expression levels of these methylated transcripts are in part regulated by the ratio of activating to repressing MBD complexes on chromatin. Consistent with this, shifting the ratio towards activation by ectopically recruiting the MBD7 complex to MBD5/6-bound loci was sufficient to partially overcome MBD5/6-mediated silencing and cause mild transcriptional upregulation and DNA demethylation. This work also highlights the power of using single-nucleus simultaneous transcriptome and methylome sequencing in plants to probe methylation dynamics at a high resolution, revealing novel antagonistic interactions between the MBD5/6 and MBD7 complexes.

## Results

### Loss of *MBD7* partially rescues the *mbd5/6* transcriptional derepression phenotype

To investigate the relationship between MBD5, MBD6, and MBD7, we generated an *mbd5/6/7* triple mutant by CRISPR/Cas9 (Supplementary Fig. 1a) and performed RNA-seq in mature pollen (Supplementary Data 1). Using our recently updated pollen transcriptome annotation[6], we identified 202 significantly upregulated transcripts in *mbd5/6* pollen (Fig. 1a, Supplementary Data 2), including genes, transposons, and novel uncharacterized transcripts. This list was highly consistent between two independent experiments and with published data[6] (Supplementary Fig. 1b, c). Nearly all of the *mbd5/6* upregulated transcripts were either transposons (TEs) or had TE-like DNA methylation patterns (Supplementary Fig. 2a–c), consistent with MBD5/6 primarily regulating methylated loci. This suggests that *MBD5* and *MBD6* primarily function to repress TEs in pollen. Only 37 significantly differentially expressed transcripts were found in the *mbd7* mutant, most of which (*n* = 29) were downregulated (Fig. 1a). This is consistent with the previously described role of MBD7 as an inhibitor of silencing with few endogenous targets[15–18].

Upregulation of many *mbd5/6* upregulated transcripts was rescued in the *mbd5/6/7* triple mutant (Fig. 1a–c). Using hierarchical clustering of TPM estimates across our pollen RNA-seq dataset (Fig. 1d), we classified the *mbd5/6* upregulated transcripts based on extent of rescue: 101 (50%) transcripts were mostly restored to wild-type expression levels in *mbd5/6/7* ("rescued"), 35 (17%) transcripts were partially restored ("partially rescued"), and 53 (26%) transcripts had similar expression levels in *mbd5/6* and *mbd5/6/7* ("not rescued") (Fig. 1c–e, Supplementary Fig. 1d). An additional 13 *mbd5/6* upregulated transcripts that did not fall cleanly into these three categories

were omitted from this analysis. Most *mbd5/6* upregulated transcripts were not misregulated in *mbd7* alone (Fig. 1d). The few significantly downregulated transcripts in *mbd7* were generally only mildly downregulated in *mbd7*, strongly upregulated in *mbd5/6*, and rescued in *mbd5/6/7* (Fig. 1b, Supplementary Fig. 1e). This suggests that MBD7 primarily promotes the expression of MBD5/6 targets, and that MBD7 can activate these targets much more strongly in *mbd5/6* mutants than in wild-type. Together, our data indicate that MBD5/6 and MBD7 regulate an overlapping set of transcripts in pollen, and function antagonistically at these loci.

### MBD5/6 and MBD7 prefer different subsets of CG-methylated loci

To investigate the interaction between MBD5/6 and MBD7 at target loci, we performed ChIP-seq of three independent lines of flag-tagged MBD7 in *mbd7* (MBD7-Flag) alongside two independent lines of flag-tagged MBD6 in *mbd6* (MBD6-Flag) (Supplementary Fig. 3a, Supplementary Data 1) using unopened flower buds. Since MBD5 and MBD6 are part of the same complex, they highly colocalize by ChIP-seq[4], and our MBD6-Flag ChIP-seq data were consistent with published MBD5 and MBD6 ChIP-seq data[4] (Supplementary Fig. 3b). Additionally, both MBD6 and MBD7 were correlated with DNA methylation density with a strong preference for CG over non-CG methylation, consistent with previous reports[4,18], (Supplementary Fig. 3c).

Since MBD6 and MBD7 showed a shared preference for regions with high CG methylation (mCG) density, we predicted that MBD6 and MBD7 would bind similar sites in the genome. We quantified this by identifying peaks of MBD6 and MBD7 occupancy and then measuring average ChIP-seq signal for all samples across the union of all MBD6 and MBD7 peaks (Fig. 2a, b). As predicted, MBD6 and MBD7 occupied a strongly overlapping set of sites in wild-type, with similar preference for high mCG density (Fig. 2a, b, Supplementary Fig. 3c). However, while both MBD6 and MBD7 were enriched at most of these regions, they were also anticorrelated: the peaks with the highest MBD6 occupancy tended to have the lowest MBD7 occupancy and vice-versa (Fig. 2a, b). This suggests that while both MBD5/6 and MBD7 have affinity for high mCG density, other factors help fine-tune their localization.

To understand these differences in MBD5/6 and MBD7 occupancy, we first explored the subcellular localization of these proteins. In roots, which are highly amenable to confocal live cell imaging of fluorescently-tagged proteins, MBD5 and MBD6 form strong nuclear foci that depend on interactors ACD15 and ACD21 and overlap chromocenters[5]. MBD7 also interacts with a pair of ACD proteins, IDM3 and IDM2, and can form nuclear foci in vivo when overexpressed[16,24]. We therefore investigated the interaction between fluorescently-tagged MBD6 and MBD7 in roots. We first confirmed that MBD7 fused to YFP also formed nuclear foci in roots when expressed from its native promoter (Fig. 2c). Unlike MBD6[5], MBD7 foci only partially overlapped chromocenters and were only partially abolished in the absence of the MBD7 ACD protein interactor IDM3 (Fig. 2c, Supplementary Fig. 3d, e). We next co-expressed MBD6-RFP and MBD7-YFP in wild-type plants. While MBD6 and MBD7 foci often overlapped, as expected given their similar affinity for mCG density (Supplementary Fig. 3c) and overlapping occupancy by ChIP (Fig. 2a), MBD6 and MBD7 foci were much less likely to overlap than a control co-expressing MBD6-RFP and MBD6-YFP (Fig. 2c, Supplementary Fig. 3f). This is consistent with MBD5/6 and MBD7 having distinct binding patterns despite a shared set of target sites.

We next focused on three groups of MBD6/7 shared peaks by ChIP-seq: the top 10% of peaks bound by MBD7, which tended to have lower MBD6 occupancy (MBD7-dominant peaks), the bottom 10% of peaks bound by MBD7, which had high MBD6 occupancy (MBD6-dominant peaks), and the middle 10% of peaks, which are bound by both MBD6 and MBD7 but with intermediate enrichment (mixed

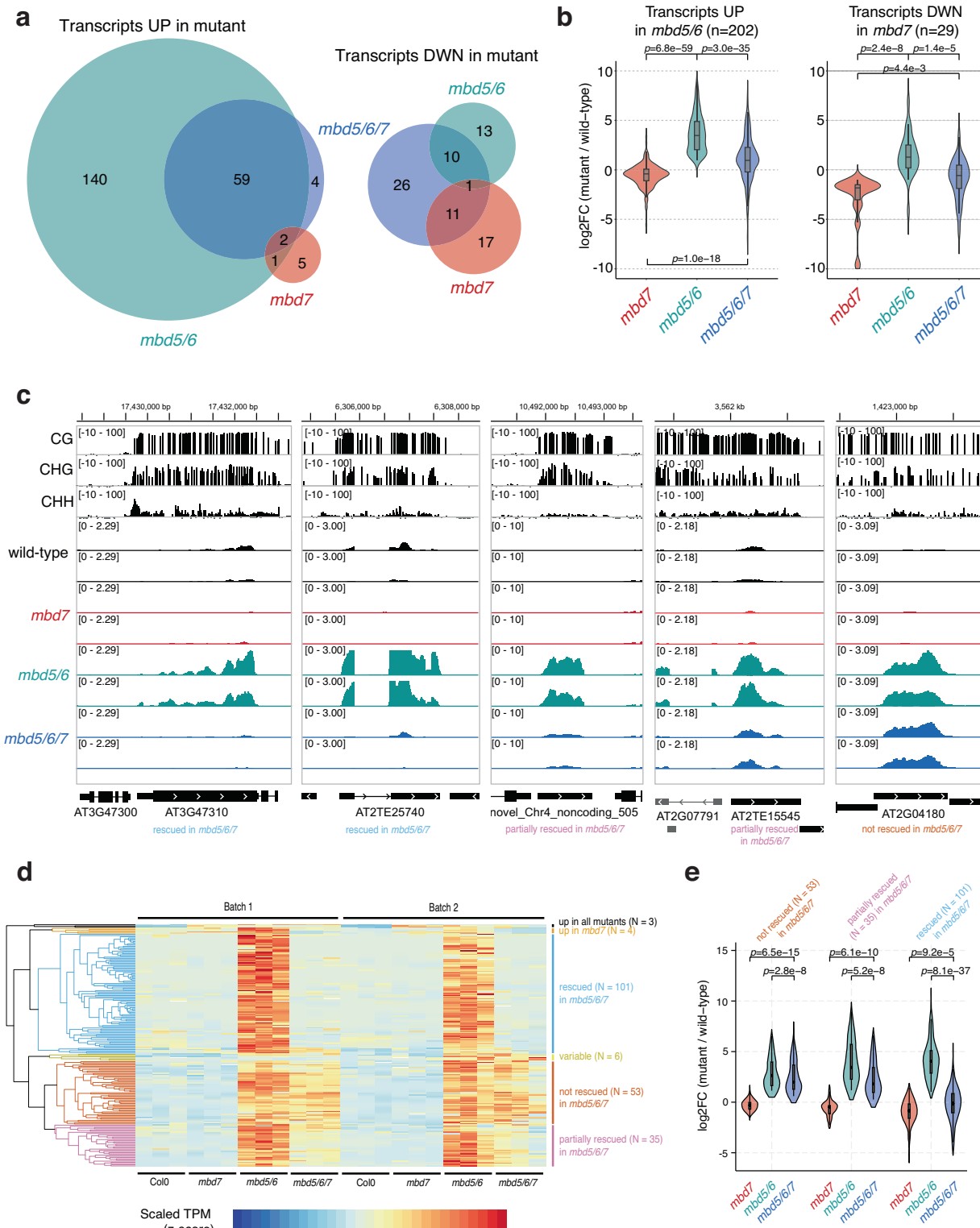

**Fig. 1 | MBD7 activates loci repressed by MBD5 and MBD6 in pollen. a** VENN diagrams showing the intersection of significantly upregulated and down-regulated transcripts in mature pollen by RNA-seq in indicated mutants. **b** Distribution of expression change (log$_2$ fold-change in indicated mutant over wild-type control, $n = 6$ biological replicates per genotype) in mature pollen for the indicated groups of differentially expressed transcripts ($n = 202$ upregulated transcripts, $n = 29$ downregulated transcripts). *P*-value: two-sided paired *t*-test. **c** Examples of genes repressed by MBD5/6, and either rescued, partially rescued, or not rescued by loss of MBD7 (see **d**). Upper tracks: wild-type whole-genome bisulfite-seq in mature pollen. Lower tracks: RNA-seq in mature pollen from two independent experiments, average of 3 biological replicates each. **d** Heatmap of pollen RNA-seq expression for the 202 transcripts upregulated in *mbd5/6*. TPM: transcripts per million, scaled by row. **e** Like B, but over *mbd5/6* upregulated transcripts rescued ($n = 101$), partially rescued ($n = 35$), or not rescued ($n = 53$) by *mbd7*. The log$_2$ fold-change is calculated over $n = 6$ biological replicates per genotype. **b, e** Boxplots within violin plots report the median (central line), the 25th (Q1) and 75th (Q3) percentiles (box edges), and largest/smallest value within 1.5 * interquartile range (whiskers). *P*-value: two-sided paired *t*-test.

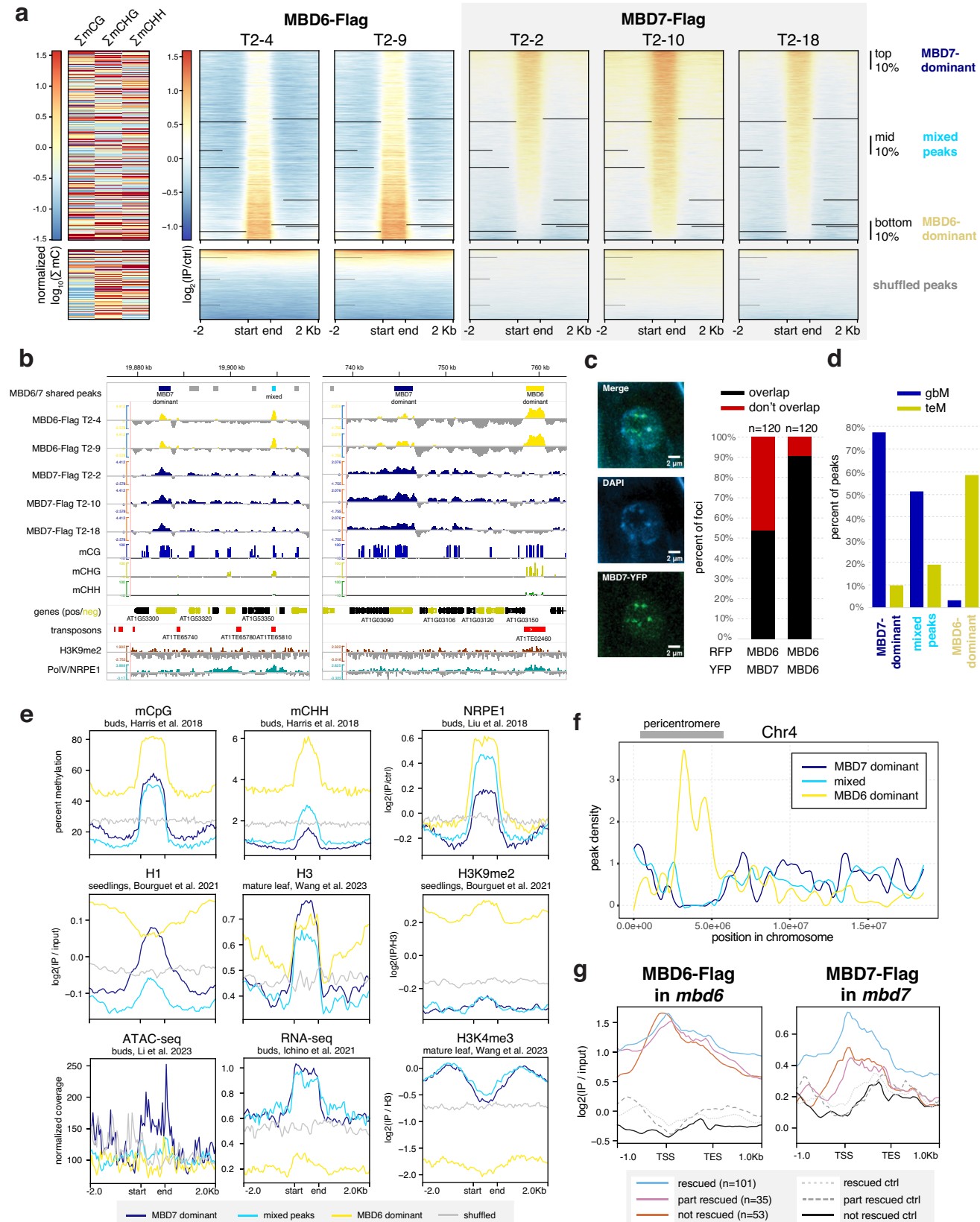

**a** MBD6-Flag / MBD7-Flag panels (T2-4, T2-9, T2-2, T2-10, T2-18) with MBD7-dominant (top 10%), mixed peaks (mid 10%), MBD6-dominant (bottom 10%), and shuffled peaks.

**b** Genome browser tracks: MBD6/7 shared peaks, MBD6-Flag T2-4, MBD6-Flag T2-9, MBD7-Flag T2-2, MBD7-Flag T2-10, MBD7-Flag T2-18, mCG, mCHG, mCHH, genes (pos/neg), transposons, H3K9me2, PolV/NRPE1.

**c** Merge, DAPI, MBD7-YFP confocal images; overlap/don't overlap bar chart.

**d** gbM/teM percent of peaks across MBD7-dominant, mixed peaks, MBD6-dominant.

**e** Metaplots: mCpG, mCHH, NRPE1, H1, H3, H3K9me2, ATAC-seq, RNA-seq, H3K4me3 for MBD7 dominant, mixed peaks, MBD6 dominant, shuffled.

**f** Chr4 peak density along chromosome position for MBD7 dominant, mixed, MBD6 dominant.

**g** MBD6-Flag in *mbd6* and MBD7-Flag in *mbd7* metaplots: rescued (n=101), part rescued (n=35), not rescued (n=53), rescued ctrl, part rescued ctrl, not rescued ctrl.

peaks) (Fig. 2a, b). Comparing these peaks to published inflorescence DNA methylation data[25], we found that all three sets of peaks were enriched over regions with elevated DNA methylation (Fig. 2a, b, d, e), as expected given that both MBD6 and MBD7 binding correlates with mCG density (Supplementary Fig. 3c). However, while CG methylation density at MBD7-dominant peaks was very similar to MBD6-dominant peaks, CHG and CHH methylation (mCHG and mCHH, respectively) was largely absent in MBD7-dominant peaks (Fig. 2a, b, d, e). This was reminiscent of 'gene body methylation' (gbM), which commonly occurs over genes and is characterized by elevated mCG alongside

**Fig. 2 | MBD5/6 and MBD7 prefer different subsets of CG-methylated loci.**
**a** (Left) Average methylation density (sum of % methylation across all cytosines in indicated sequence context) in 400 bp windows centered on the midpoint of the peaks shown at right. Each column was normalized separately. (Right) Heatmaps of MBD6-Flag signal and MBD7-Flag signal (log$_2$ IP over no-FLAG control) over the union of all MBD6 and MBD7 peaks. Each heatmap is an independent transgenic line (T2-[line number]). All heatmaps in (**a**) share same row order. **b** Browser tracks showing the location of MBD6/7 union peaks (top), as well as MBD6 and MBD7 ChIP-seq signal (log$_2$ IP over no-FLAG control), % DNA methylation data from inflorescences with unopened flower buds[25], gene annotations (black = forward strand, yellow = reverse), and transposon (TE) annotations. Genes and TEs are from the araport11 annotation[61]. Both Col replicates from ref. 25 were pooled; only sites with 5 or more coverage shown; small negative value (−10%) corresponds to sites with coverage but no methylation, to distinguish from missing data. H3K9me2[63] and PolV (NRPE1)[56] ChIP-seq signal also shown. **c** (left) Representative image of root nucleus expressing pMBD7::MBD7-YFP and incubated with DAPI to stain

chromocenters. (right) Percent of foci assayed ($n = 120$ for each condition) where YFP and RFP signal overlapped vs. did not overlap. Left bar shows co-expression of MBD6-RFP and MBD7-YFP, right bar shows control co-expressing MBD6-RFP with MBD6-YFP. **d** Percent of MBD7-dominant, mixed, and MBD6-dominant peaks from (A) overlapping annotated transcripts classified as either gene body methylated (gbM, mCG only) or TE-like methylated (teM, mC in all contexts). Annotations were from ref. 6. **e** Metaplots of DNA methylation[25], NRPE1[56], histone H1 and H3K9me2[63], histone H3 and H3K4me3[64], accessibility by ATAC-seq[65], and RNA-seq[4], over the peaks identified in (**a**). **f** Loess-smoothed distribution of MBD7-dominant, MBD6-dominant, and mixed peaks across Chromosome 4. Pericentromere shown as a grey bar. **g** Metaplots of MBD6 and MBD7 ChIP-seq log$_2$(IP/no-FLAG control) signal over pollen *mbd5/6* upregulated transcripts rescued, partially rescued, and not rescued by MBD7 loss (see Fig. 1d). Controls are an equal number of random genes matched for expression level and length. Transgenic lines were averaged ($N = 2$ for MBD6-Flag, $N = 3$ for MBD7-Flag).

absence of mCHG and mCHH[26]. By contrast, MBD6-dominant peaks tended to be in regions strongly methylated in all sequence contexts, which is more common over silent TEs and in deep heterochromatin (Fig. 2a, b, d, e). Consistent with this, MBD6-dominant peaks were highly enriched in the pericentromere, which is mostly silent heterochromatin, whereas MBD7-dominant and mixed peaks were strongly depleted from the pericentromere and enriched in the chromosome arms, which are gene rich (Fig. 2f). MBD6-dominant peaks were also associated with very high levels of H3K9me2, but low H3K4me3, accessibility, and transcription (Fig. 2e). MBD7-dominant peaks instead were much more likely to be in or near accessible chromatin, characterized by elevated DNA accessibility, H3K4me3, H3K27ac, and transcription, and depleted H3K9me2 (Fig. 2e, f, Supplementary Fig. 4a). However, MBD7-dominant peaks were strongly enriched for H1 directly over the peak, despite peaks occurring in regions that are broadly depleted for H1, whereas MBD6-dominant peaks were in regions of broad H1 enrichment with local depletion of H1 over the peak (Fig. 2e). Together, these data indicate that MBD6 preferentially binds regions of very high DNA methylation within broad heterochromatic domains, while MBD7 prefers euchromatic gene body methylation. This is broadly consistent with the MBD5/6 complex functioning in transcriptional repression and the MBD7 complex in transcriptional activation.

There are multiple possible explanations for the difference in MBD6 and MBD7 occupancy. One possibility is that the MBD5/6 complex affects MBD7 binding directly, potentially by occupying sites that could otherwise be bound by MBD7. To investigate this, we performed ChIP-seq of MBD7-Flag transformed into both the *mbd5/6/7* and *mbd7;sln* mutant backgrounds (MBD7-Flag *mbd5/6* and MBD7-Flag *sln* respectively) (Supplementary Fig. 3a). Both *mbd5/6* and *sln* disrupt MBD5/6 complex function, but in *sln* MBD5 and MBD6 are still available to bind to their DNA targets[4]. Across two independent MBD7-Flag *mbd5/6* lines and three independent MBD7-Flag *sln* lines, we saw no substantial difference in MBD7 affinity for mCG density (Supplementary Fig. 3c), or occupancy over either *mbd7* downregulated transcripts (Supplementary Fig. 4b), MBD6/7 union peaks (Supplementary Fig. 4c), or *mbd5/6* upregulated transcripts (Supplementary Fig. 4d). This suggests that the MBD5/6 complex itself does not directly affect MBD7 localization. Instead, we hypothesize that MBD7 cannot easily penetrate deep heterochromatin, whereas MBD5/6, as a repressive complex, may be able to access these sites more easily.

We next examined MBD6 and MBD7 occupancy over loci upregulated in *mbd5/6* pollen. All *mbd5/6* upregulated loci, regardless of rescue by *mbd7*, were strongly enriched for MBD6 over the entire locus and particularly at the TSS (Fig. 2g, Supplementary Fig. 4d), suggesting these loci are mostly direct MBD5/6 targets. However, MBD7 occupancy over these transcripts was variable: while all *mbd5/6* upregulated

transcripts showed some MBD7 enrichment compared to control transcripts, particularly at the TSS, *mbd7* "rescued" transcripts (Fig. 1d) were substantially more likely to be bound by MBD7 than genes that were only partially or not at all rescued by loss of MBD7 (Fig. 2g, Supplementary Fig. 4d). MBD6 occupancy was also lower over *mbd7* downregulated loci than over *mbd5/6* upregulated loci, consistent with the general anticorrelation between MBD6 and MBD7 binding (Supplementary Fig. 4e). We concluded that although most sites strongly bound by MBD5/6 are only weakly bound by MBD7 and vice-versa (Fig. 2a), a subset of MBD5/6 targets are also strongly bound by MBD7, and these are upregulated by MBD7 in the absence of *mbd5/6*.

## Single-nucleus multi-omics reveals DNA methylation dynamics in developing pollen nuclei

MBD7 has been proposed to enhance the expression of methylated loci by facilitating active DNA demethylation[16–18]. To understand how MBD5/6 and MBD7 are interacting at their shared targets, we sought to test whether MBD7 promotes DNA demethylation at *mbd5/6* upregulated transcripts in pollen. No loss of DNA methylation in *mbd5/6* was previously detected in whole floral tissue[4]. However, the transcriptional derepression in *mbd5/6* occurs primarily in the VN of pollen[6], suggesting the same could be true of DNA methylation changes. To examine the effect of loss of MBD5/6 on DNA methylation in pollen nuclei at high resolution, we performed simultaneous single-nucleus methylcytosine and transcriptome profiling of single pollen nuclei (snmCT-seq)[27] (Fig. 3a). We obtained a total of 8284 pollen nuclei across two independent experiments, 4150 from Col and 4134 from *mbd5/6* (Supplementary Data 1, 3). Using Seurat[28], we identified clusters of nuclei based on transcriptional patterns and assigned cluster identities based on known marker genes[6,29–31] (Fig. 3b, c, Supplementary Fig. 5a, b, Supplementary Data 3). This analysis identified all the expected pollen cell types[32] and was consistent with clusters that we had previously identified by 10x snRNA-seq in pollen[6]: microspore nuclei (MN), generative nuclei (GN), sperm nuclei (SN), and a progression of clusters of vegetative nuclei (VN), spanning from immediately post-mitosis I (VN1) to mature (VN5) (Fig. 3b, c, Supplementary Fig. 5c). Two additional clusters with intermediate expression patterns when compared to the published dataset were named "MN to VN" and "VN1 to 2" based on the two most similar published clusters (Fig. 3b, c, Supplementary Fig. 5c). The VN clusters formed a clear developmental trajectory from microspores (MN) transitioning into VN (MN to VN), to immature VN (VN1-VN3) to mature VN (VN4-VN5), as previously observed[6] (Fig. 3b, arrow). MBD7 was highly and specifically expressed in immature VN (primarily clusters VN1, VN1to2), as was DME, consistent with reports that DME is required for demethylation in VN (Supplementary Fig. 5d)[13]. MBD5 and MBD6, while more lowly expressed, were also enriched in immature VN (Supplementary Fig. 5d, e). Linker histone H1 has also been reported to be depleted in the VN,

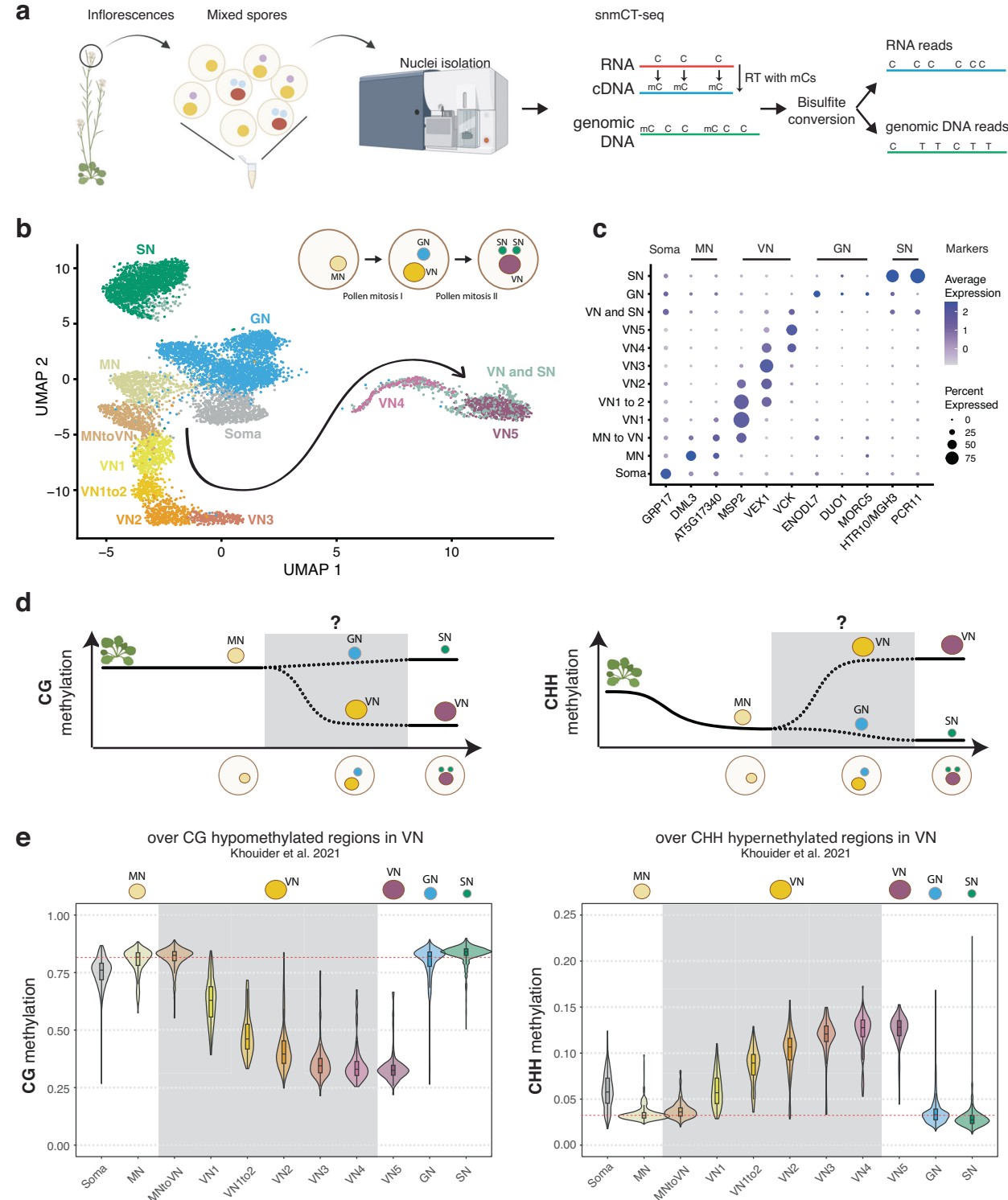

which is thought to help promote DNA demethylation and loss of chromatin condensation[11]. Consistent with this, in both our previous snRNA-seq data[6] and in the snmCT-seq dataset, H1 expression is present in the MN but largely absent from VN clusters (Supplementary Fig. 5f). Expression of the MBDs, H1, DME, and other DNA methylation-related genes examined was not substantially affected in *mbd5/6* (Supplementary Fig. 5d–f). Overall, the transcriptome data from snmCT-seq was highly consistent with previous 10x single nuclei data[6], highlighting the robustness of both methods for detecting different nuclei types in pollen.

Next, we inspected DNA methylation levels in pollen nuclei in wild-type (Supplementary Data 4). A preliminary dimensionality reduction analysis of the snmCT-seq methylation data alone, aggregated across 25 kb bins genome-wide, showed clear separation of VN nuclei from the other nuclei types, suggesting that the VN adopts a particularly unique methylation state (Supplementary Fig. 6a–c), consistent with prior reports[7,12,13]. Other nuclei types were generally not well separated, suggesting single-nuclei methylomes alone are insufficient to identify most pollen nuclei types, at least with the level of resolution possible from low-coverage single-nucleus data. Based

**Fig. 3 | snmCT-seq reveals DNA methylation dynamics in developing pollen nuclei. a** Overview of pollen isolation and snmCT-seq protocol (see methods)[6,27], created in BioRender. Picard, C. (2025). https://BioRender.com/qutlii3. **b** UMAP of pollen nuclei, showing developmental trajectories from microspore nuclei (MN) into sperm nuclei (SN) and vegetative nucleus (VN). A subset of likely doublets of VN and SN, which may stick together during sorting, are separated into their own cluster ("VN and SN") (see methods, Supplementary Fig. 12). **c** Expression of marker genes for different pollen nuclei types, previously identified by 10x [6]; most of these markers have also been independently validated in other studies. The three VN markers are for early post-mitotic (MSP2), mid-stage (VEX1) and late-stage (VCK) VN. **d** Diagram of approximate relative DNA methylation levels during development from vegetative tissue to MN, and finally to mature SN and VN. Grey box shows dynamics revealed in this study. Published data from FACS-sorted populations of MN and mature VN and SN were used for other parts of the diagram[7,12,13], and are also consistent with our data. Rosette image is from BioRender. Picard, C. (2025) https://BioRender.com/vciw129. **e** Distribution of average per-nucleus snmCT-seq methylation for nuclei assigned to each cluster in (**b**). Average methylation was computed over regions hypomethylated in the CG context in VN (VN CG hypo DMRs, left) and hypermethylated in the CHH context in VN (VN CHH hyper DMRs, right). These regions were previously identified from bisulfite-sequencing of purified FACS-sorted VN populations[13]. The red horizontal line marks the median methylation level in MN, for reference. The number of nuclei in each cluster/genotype used to create the violin plot distributions shown are in Supplementary Data 3 (tab 'nuclei per cluster', expt 1 and 2 pooled). Boxplots within violin plots report the median (central line), the 25th (Q1) and 75th (Q3) percentiles (box edges), and largest/smallest value within 1.5 * interquartile range (whiskers).

on published bisulfite sequencing of FACS-purified populations of pollen nuclei, MN methylation levels largely resemble somatic nuclei but are decreased in the CHH context (Fig. 3d)[12]. The methylome of SN resembles MN, with a small increase in CG methylation and decrease in CHG and CHH methylation (Fig. 3d, Supplementary Fig. 7a)[7,12,13]. The mature VN is strongly hypermethylated in the CHH context relative to MN, with minor CHG methylation increases in the pericentromere, while simultaneously undergoing some loss of CG and CHG methylation due to active demethylation by DME or ROS1 (Fig. 3d, Supplementary Fig. 7a)[7,12,13]. However, these published datasets are from many pooled cells that were purified based on limited information, and lack the resolution of single-cell approaches. Moreover, methylation data for GN and immature VN, the time points between pollen mitosis I and II, has not yet been generated (Fig. 3d, Supplementary Fig. 7a). We therefore used our snmCT-seq dataset to examine DNA methylation dynamics during wild-type pollen development in more detail, using our RNA expression-based clustering (Fig. 3b).

We first generated chromosome-wide methylation profiles of nuclei in different transcriptionally defined clusters to observe global methylation behavior during pollen development (Supplementary Fig. 7b). As expected, global CG methylation levels were largely unchanged across clusters, with a small decrease in the VN lineage and increase in SN relative to GN and MN (Supplementary Fig. 7b). This is consistent with previous studies which found CG hypermethylation in the SN[7,12,33]. Our data indicates that this occurs after pollen mitosis II, since it is not apparent in the GN cluster (Supplementary Fig. 7b). Pericentromeric CHG methylation was mildly increased in the VN lineage relative to MN, but decreased in SN relative to GN (Supplementary Fig. 7b), also consistent with previous reports[7,12,33]. Interestingly, the CHG methylation increase relative to MN was apparent as early as in the MNtoVN cluster, and remained mostly stable throughout VN development (Supplementary Fig. 7b). Pericentromeric CHH methylation strongly increased in the VN lineage as expected from previous studies[7,12,33], and this methylation gain appeared to be progressive, occurring gradually along the VN maturation trajectory (Supplementary Fig. 7b).

We next examined methylation patterns in our wild-type nuclei over previously identified regions with less CG methylation in VN compared to SN ("CG hypo DMRs", which are likely DME targets) and more CHH methylation in VN compared to SN (CHH hyper DMRs) based on FACS-sorted populations[13]. As expected[7,12,13], CG methylation over CG hypo DMRs was dramatically decreased in the wild-type VN nuclei, whereas CHH methylation over CHH hyper DMRs decreased in MN, remained low in GN and SN, and greatly increased in the VN (Fig. 3d, e, Supplementary Fig. 7c)[13]. CHG methylation levels at CG hypo DMRs were also mildly decreased, despite the increase observed at pericentromeres (Supplementary Fig. 7b, c). This is consistent with the reported in vitro activity of DME on cytosine methylation in any sequence context[34]. DME was previously reported to mainly target TEs in the chromosome arms in the vegetative cell[13]. Indeed, loss of CG methylation in VN was strongest at euchromatic TEs, consistent with

DME targeting patterns[7,12,13], while CHH hypermethylation occurred mainly at heterochromatic TEs (Supplementary Fig. 7d). Both loss of CG methylation and gain of CHH methylation in VN also occurred over MBD6 ChIP-seq peaks, suggesting that MBD5/6 bind targets of DNA methylation reprogramming in the VN (Supplementary Fig. 7e).

The increased resolution of our single-cell dataset also allowed us to interrogate the relative timing of methylation changes in VN. Methylation loss, presumably due to DME activity[7,12,13], was already apparent in the VN1 cluster, which in pseudotime represents the 'earliest' non-MN population of nuclei along the VN developmental trajectory, and decreased further in both VN1to2 and VN2 nuclei before mostly plateauing (Fig. 3e, Supplementary Fig. 7b–e). By definition, transcriptional changes occurred throughout the progression of MN to VN5 nuclei, since that is how the clusters were identified. However, our data shows that DNA methylation loss due to DME activity plateaus by VN3. This suggests that active removal of CG methylation in VN by DME and/or ROS1[13] primarily occurs immediately after pollen mitosis I, rather than gradually throughout VN maturation. Consistent with this trend, DME was also highly expressed in immature VN (VN1 and VN1to2) but decreased rapidly by VN3 (Supplementary Fig. 5d)[6,8]. Similarly, the CHH methyltransferase CMT2, while lowly expressed overall, was more highly expressed in immature VN, and CHH methylation increased more rapidly in immature VN compared to mature VN (Fig. 3e, Supplementary Fig. 5d). Thus, our data both confirmed existing observations of pollen methylation dynamics and provided insights into the timing of key DNA methylation changes during pollen development.

## mbd5/6 mutants lose additional methylation in immature VN, preceding MBD5/6 target gene upregulation

MBD5 and MBD6 are primarily expressed in immature VN (Supplementary Fig. 5d)[6] which is also where we observed the biggest VN methylation changes in wild-type (Fig. 3e), suggesting a potential link between MBD5/6 and DNA methylation reprogramming in immature VN. Consistent with this, our MBD6 ChIP-seq data showed substantial overlap between MBD6-bound regions in inflorescences and regions that are actively demethylated by DME or ROS1 in the VN (CG hypo DMRs)[13] (Fig. 4a, Supplementary Fig. 8a). To test whether loss of MBD5/6 affects VN methylation dynamics, we used our snmCT-seq data to compare methylation levels in mbd5/6 and wild-type in the different pollen nuclei clusters. We observed a marked loss of CG and CHG methylation specifically in the mbd5/6 VN relative to wild-type, along with a minor loss of methylation in the other pollen nuclei types (Fig. 4b–d, Supplementary Fig. 8b). Methylation loss in mbd5/6 relative to wild-type was most pronounced at MBD6 ChIP-seq peaks overlapping VN CG hypo DMRs (Fig. 4c, Supplementary Fig. 8b), suggesting that MBD5/6 prevent excessive demethylation by DME and/or ROS1 at these sites. A smaller but clear loss of CG and CHG methylation also occurred at MBD6 peaks not overlapping VN CG hypo DMRs (Fig. 4d, Supplementary Fig. 8b), as well as at VN CG hypo DMRs not overlapping MBD6 peaks (Fig. 4b, Supplementary Fig. 8b). However, VN CG

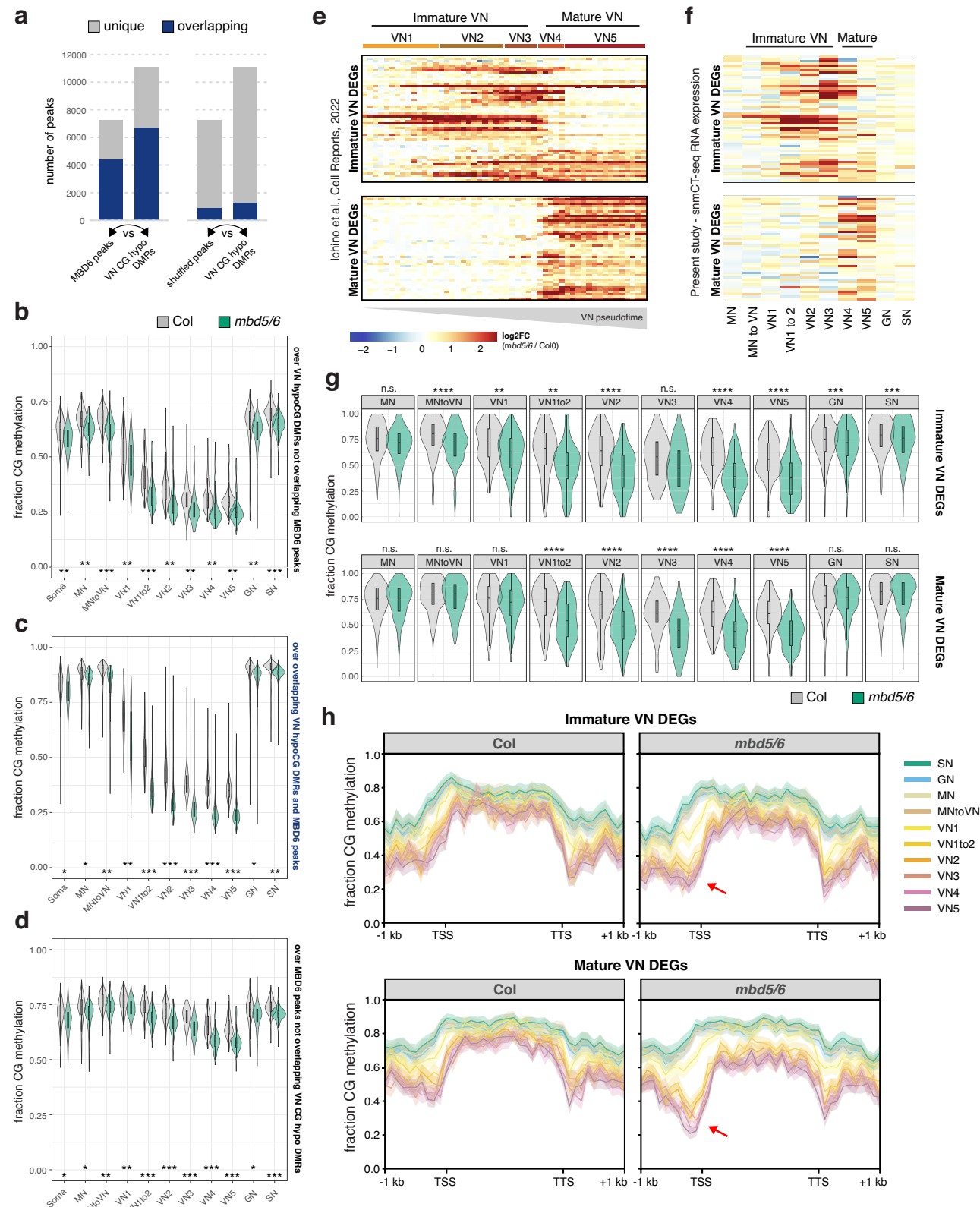

hypo DMRs that did not overlap a significant MBD6 peak were still consistently mildly enriched for MBD6 compared to surrounding regions (Supplementary Fig. 8a), which may explain why we still observed hypomethylation at these sites in *mbd5/6* (Fig. 4b, Supplementary Fig. 8b). We observed only minor loss of methylation in *mbd5/6* genome-wide or at VN CHH hyper DMRs (Supplementary Fig. 8c, d), suggesting that loss of MBD5/6 primarily affects sites

undergoing active demethylation in VN (CG hypo DMRs[13]). There were no major changes in the expression level of DME or other key DNA methylation-related genes in *mbd5/6* (Supplementary Fig. 5d, e), and the most strongly CG-demethylated sites were bound by MBD6 in inflorescence ChIP-seq (Fig. 4a, c), suggesting that the methylation changes in *mbd5/6* were likely directly caused by loss of MBD5/6 binding at their targets. Additionally, *mbd5/6* VN becomes most

**Fig. 4 | *mbd5/6* mutants lose methylation in the immature VN, preceding MBD5/6 target gene upregulation. a** Number of MBD6 ChIP peaks that overlap VN CG hypo DMRs[13], and vice-versa. Right shows same analysis but using shuffled MBD6 peaks. Within each plot, left column shows the number of total MBD6 (true or shuffled) peaks, plus the proportion of these that overlap CG hypo DMRs. Right column shows the number of CG hypo DMRs, plus the proportion of DMRs overlapping peaks. Considered overlapping if either peak/DMR was >50% covered by the other. Distribution of per-nucleus average CG methylation levels calculated over (**b**) VN CG hypo DMRs not overlapping MBD6 peaks, (**c**) VN CG hypo DMRs overlapping MBD6 peaks, and (**d**) MBD6 peaks not overlapping VN CG hypo DMRs (regions indicated at right of each plot). Shows Col vs. *mbd5/6* CG methylation levels across all nuclei in the each snmCT-seq cluster. Stars indicate effect size of difference between WT and *mbd5/6*, Cohen's d. n.e.=no/minimal effect ( | d | <0.2), *=|d|>0.2, **=|d|>0.5, ***=|d|>0.9. **e** Heatmap of average expression in the pollen 10x dataset [6] of genes upregulated in *mbd5/6* compared to Col, clustered based on the timing of their activation with respect to the VN pseudotime trajectory (x-axis).

**f** Expression of the immature and mature VN differentially expressed loci from (**e**) in the current study's snmCT-seq data (average of nuclei from two replicates). Heatmap row order same as in (**e**). **g** Distribution of per-nucleus CG methylation levels over +/−400bp surrounding the TSSs of immature and mature VN differentially expressed loci defined in E and F. ****p-val<0.00001,***p-val<0.0001,**p-val<0.001,*p-val<0.1,n.s.=not significant, two-sided t-test comparing Col and *mbd5/6* means within indicated cluster. **h** Metaplots of average methylation profiles for Col (left) and mbd5/6 (right) over immature and mature VN differentially expressed loci across the different snmCT-seq clusters. Average of all nuclei in indicated cluster across two snmCT-seq replicates, shaded region indicates standard error (SE). **b**–**d**, **g**) Boxplots report the median (central line), the 25th (Q1) and 75th (Q3) percentiles (box edges), and largest/smallest value within 1.5*interquartile range (whiskers). The number of nuclei in each cluster/genotype used to create the violin plot distributions shown are in Supplementary Data 3 (tab 'nuclei per cluster', expt 1 and 2 pooled).

strongly demethylated relative to Col at the loci (VN CG hypo DMRs) and in the nuclei types (immature VN) that are already undergoing active demethylation by DME and/or ROS1 in wild-type (Fig. 3e, Fig. 4a–d, Supplementary Fig. 5d). This suggests that MBD5/6 helps protect their bound targets from the wave of active demethylation that occurs during VN development.

We next compared these methylation changes to the transcriptional changes in *mbd5/6*, which also occur primarily in VN[6]. Using our previous 10x snRNA-seq dataset[6], we identified transcripts that become upregulated in *mbd5/6* beginning in either immature VN (VN1-3) or in mature VN (VN4-5) (Fig. 4e). These transcripts showed the same immature/mature expression pattern in our snmCT-seq data (Fig. 4f). At transcripts upregulated in immature *mbd5/6* VN, *mbd5/6* nuclei showed a significant drop in average CG methylation relative to Col over the TSS-proximal region (+/− 400 bp around the TSS) beginning in the "MN to VN" cluster (Fig. 4g, h, Supplementary Fig. 8e–g), preceding the earliest activation of these genes (Fig. 4e, f). Similarly, transcripts upregulated in mature *mbd5/6* VN showed significant CG methylation loss relative to Col by the "VN1 to 2" cluster (Fig. 4g, h, Supplementary Fig. 8e, g), again preceding the activation of these genes in pseudotime, which occurs in the "VN4" cluster (Fig. 4e, f). Like the transcriptional upregulation, the methylation loss in *mbd5/6* at upregulated targets was specific to the VN and minimal in MN, GN, or SN (Fig. 4g, h, Supplementary Fig. 8e–g). While it's possible that the additional demethylation of *mbd5/6* upregulated transcripts is an indirect consequence of transcriptional activation, the observation that both immature and mature VN differentially expressed transcripts show evidence of demethylation in immature VN, and thereby prior to detectable gene upregulation in *mbd5/6* (Fig. 4e–h) argues against this possibility. Instead, these data suggest that in the absence of MBD5/6, the promoters of *mbd5/6* upregulated transcripts undergo additional active demethylation in VN, facilitating their expression.

**Loss of MBD7 rescues demethylation of a subset of loci in *mbd5/6* pollen**

Our data so far indicate that (1) MBD7 is required for transcriptional derepression of a subset of misregulated transcripts in *mbd5/6* VN (Fig. 1), (2) MBD5/6 and MBD7 prefer to bind different types of high mCG-regions but overlap at a subset of *mbd5/6* misregulated transcripts (Fig. 2), and (3) MBD5/6 help protect sites from excessive demethylation during VN reprogramming (Fig. 4). Since MBD7 has been proposed to promote demethylation[16–18], we hypothesized that MBD5 and MBD6 help prevent excessive demethylation during VN reprogramming by antagonizing MBD7. If this model is correct, then the loss of methylation in *mbd5/6* should be rescued in *mbd5/6/7*.

We first tested this by performing BS-seq in mature pollen. We detected no major genome-wide changes in methylation in *mbd5/6* or in *sln*, which has a similar transcriptional phenotype as *mbd5/6*[4]

(Supplementary Fig. 9a), consistent with previous reports[6] and with the *mbd5/6* snmCT-seq data (Supplementary Fig. 7b). We next focused on the set of *mbd5/6* upregulated transcripts, which are demethylated around the TSS in *mbd5/6* VN relative to Col (Fig. 4h). Although the methylation loss was diluted in whole pollen, we were able to detect a small but consistent decrease in CG and CHG methylation in both *mbd5/6* and *sln* around the TSS of *mbd5/6* upregulated transcripts (Supplementary Fig. 9b–e). The hypomethylation was also detectable at MBD6 target sites that become demethylated in VN[13] (Supplementary Fig. 9c). The *mbd5/6* hypomethylation was fully rescued in *mbd5/6/7* at the *mbd5/6* upregulated transcripts that were transcriptionally rescued by loss of MBD7, but only partially at "not rescued" loci and other MBD6 peaks (Supplementary Fig. 9b–e).

While we were able to detect a mild demethylation phenotype in *mbd5/6* whole pollen, and saw rescue in *mbd5/6/7* at some loci (Supplementary Fig. 9b–e), the difference was very modest relative to our snmCT-seq data (Fig. 4c), presumably due to VN nuclei comprising only a subset of the whole pollen samples. We therefore performed an additional replicate of developing pollen snmCT-seq, this time comparing wild-type, *mbd7*, *mbd5/6* and *mbd5/6/7* together. After quality filtering, we obtained over 1000 high-quality nuclei per genotype, with 5,638 nuclei total for this replicate (Supplementary Data 3). To compare these data to our previous results, we projected the new transcriptomic data onto our previous analysis and UMAP. We again identified all the expected cell types, with expression patterns broadly consistent with our original clustering (Supplementary Fig. 10a, b, Supplementary Data 3). Compared to the first two experiments, this experiment had fewer nuclei assigned to the mature VN (VN4 and VN5) and mature sperm (SN) clusters, possibly due to differences in the ages of the plants used (Supplementary Fig. 10a, b, Supplementary Data 3). Due to this, VN4 and VN5 were combined into a single 'mature VN' cluster for this analysis.

Wild-type and *mbd5/6* methylation patterns across nuclei clusters in our third snmCT-seq experiment were highly consistent with our previous results, with comparable loss of CG methylation and gain of CHH methylation over VN CG hypo and CHH hyper DMRs[13] in wild-type VN, as well as additional loss of CG methylation at VN CG hypo DMRs in *mbd5/6* VN nuclei (Supplementary Fig. 10c). At VN CG hypo DMRs, the additional demethylation in *mbd5/6* VN was strongly, but not fully, rescued in *mbd5/6/7* VN (Fig. 5a). Interestingly, the *mbd7* single mutant showed mildly increased CG methylation levels relative to wild-type in the VN, but not in other nuclei types, suggesting that MBD7 facilitates the wave of CG demethylation during VN development (Fig. 5). We next looked at methylation patterns at transcriptionally 'rescued' loci compared to 'not rescued' loci (Fig. 1d). We found that most of the methylation loss in *mbd5/6* VN was strongly rescued in *mbd5/6/7* at transcriptionally 'rescued' loci (Fig. 5b–d, Supplementary Fig. 10d). These loci also tended to be in regions with higher overall methylation

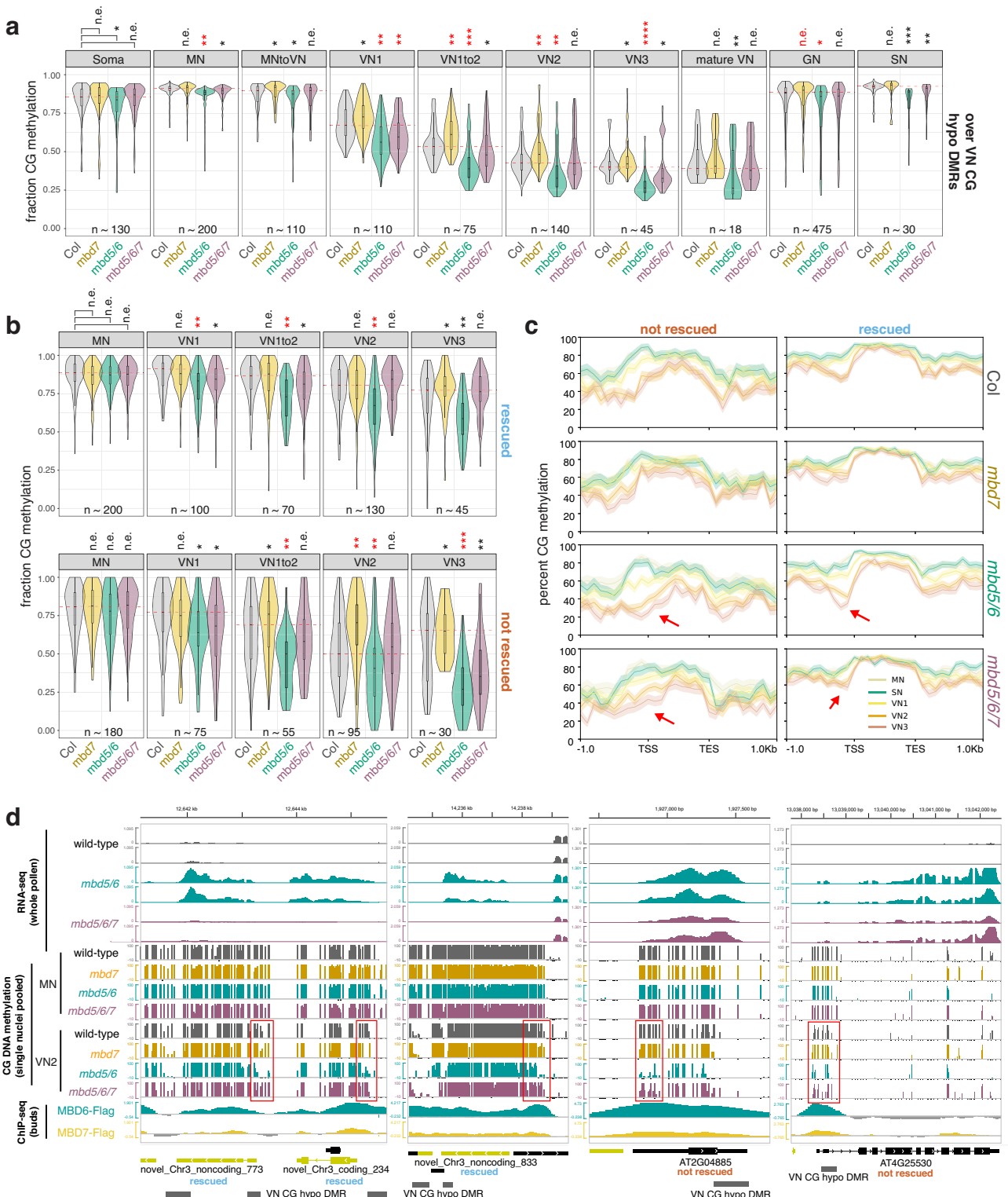

levels compared to 'not rescued' loci, and underwent minimal demethylation in wild-type VN that was restricted to the TSS-proximal region (Fig. 5b–d, Supplementary Fig. 10d). This suggests that these loci are mostly in silent chromatin that is strongly protected from active demethylation in VN. The 'not rescued' loci were generally less methylated than 'rescued' loci in wild-type, and lost substantially more methylation in wild-type VN over both the promoter/TSS region and the 5′ end of the annotated transcript (Fig. 5b–d, Supplementary Fig. 10d). Additional hypomethylation of these regions in *mbd5/6* was generally modest, and was rescued in *mbd5/6/7*, but not as

fully as at transcriptionally 'rescued' loci (Fig. 5b–d, Supplementary Fig. 10d).

No change in methylation levels was observed in *mbd7* single mutant relative to wild-type both at 'rescued' and 'not-rescued' loci (Fig. 5b–d), which is consistent with the absence of transcriptional downregulation of these loci in *mbd7* (Fig. 1d, e) and indicates that these loci cannot be demethylated by MBD7 in the presence of MBD5/6. However, the TSS region of *mbd7* downregulated transcripts showed increased CG methylation in *mbd7*, supporting the idea that MBD7 promotes DNA demethylation (Supplementary Fig. 10e), as

**Fig. 5 | Loss of MBD7 rescues excessive demethylation in *mbd5/6* pollen.**
**a** Average CG methylation across the four genotypes tested in snmCT-seq experiment 3, over VN CG hypo DMRs[13]. Each point in the violin plot represents one nucleus' average methylation over the indicated regions. A red dashed line runs through the wild-type median value. **b** Same as (A), but showing methylation averages across the TSS ( +/− 400 bp around TSS) region of *mbd5/6* upregulated transcripts 'not rescued' (bottom) or 'rescued' (top) by *mbd7*. (A-B) Number of stars indicates effect size, measured as Cohen's *d*. (n.e. = no/minimal effect ( | d | <0.2), * = |d | > 0.2, ** = |d | > 0.5, *** = |d | > 0.9, **** = |d | > 1.5). Color of stars indicates statistical significance: red color *p* < 0.001, two-sided *t*-test. **c** Metaplots of snmCT-seq rep 3 methylation data pseudobulked across all nuclei in each indicated cluster + genotype, over *mbd5/6* upregulated transcripts 'not rescued' (left) or 'rescued' (right) by *mbd7*. Average of all nuclei in indicated cluster (rep 3 only), shaded region

indicates standard error (SE). **d** Example browser images of *mbd5/6* upregulated transcripts 'not rescued' (right) or 'rescued' (left) by *mbd7*, showing RNA-seq from whole pollen (Fig. 1d), snmCT-seq CG DNA methylation data pooled by indicated genotype + cluster (only MN and VN2 clusters shown; these were selected due to the larger numbers of nuclei in these clusters, which increased the probability of getting good coverage after pooling), and ChIP-seq for MBD6 and MBD7-Flag (average of all lines) (Fig. 2). As much as possible, these loci were chosen based on having fairly even and consistent coverage in all snmCT-seq CG DNA methylation tracks shown. **a, b** Boxplots within violin plots report the median (central line), the 25th (Q1) and 75th (Q3) percentiles (box edges), and largest/smallest value within 1.5 * interquartile range (whiskers). The number of nuclei in each cluster/genotype used to create the violin plot distributions shown are in Supplementary Data 3 (tab 'nuclei per cluster', expt 3 only). *P*-value: two-sided paired t-test.

previously suggested in other tissues[16–18]. Of note, these loci are less enriched in MBD6 ChIP-seq signal compared to the *mbd5/6/7* 'rescued' loci, suggesting that MBD7 is able to demethylate them in wild-type because they are not protected by MBD5/6 (Supplementary Fig. 4e).

Taken together, these data support a model where the MBD5/6 complex protects its targets from excessive demethylation in the VN. This occurs in large part by inhibiting the MBD7 complex, which binds a subset of *mbd5/6* upregulated loci and can promote active DNA demethylation. The effect of both MBD5/6 and MBD7 becomes particularly visible during the wave of epigenetic reprogramming that occurs during VN development. However, since loss of *mbd7* only rescues the transcriptional derepression and excessive demethylation in *mbd5/6* VN at a subset of MBD5/6 loci, we hypothesize that MBD5/6 function independently of MBD7 at the remaining loci, either by directly inhibiting DNA demethylation or by inhibiting additional pathways that help facilitate demethylation during VN development.

### Recruiting the MBD7 complex to MBD5/6 targets causes mild demethylation and transcriptional upregulation

To understand the interaction between MBD5/6 and MBD7 further, we asked whether artificially recruiting excessive MBD7 complex components to MBD5/6 targets was sufficient to overcome MBD5/6 protection and cause DNA demethylation and gene upregulation. Like MBD5/6, which uses its StkyC domain to recruit its complex members, MBD7 recruits its complex via interactions between the MBD7 StkyC domain and the ACD protein IDM3[18]. Indeed, we confirmed that the MBD7 StkyC is necessary and sufficient to interact with IDM3 using BiFC (Supplementary Fig. 11a, b). We therefore created a modified MBD6 transgene where the MBD6 StkyC domain was replaced with the MBD7 StkyC domain, named MBD6^SWAP (Fig. 6a). Using BiFC, we confirmed that MBD6^SWAP interacted with IDM3, and not with ACD15 (Fig. 6b). Like wild-type MBD5/6 and MBD7, MBD6^SWAP formed nuclear foci in root nuclei (Supplementary Fig. 11c). However, while wild-type MBD6 foci were dependent on ACD15 and ACD21 as previously observed[5], MBD6^SWAP foci were fully dependent on IDM3 (Supplementary Fig. 11d). Thus, we concluded that MBD6^SWAP recruits the MBD7 complex rather than the MBD5/6 complex.

We next transformed our MBD6^SWAP construct into both wild-type (WT + MBD6^SWAP) and *mbd5/6* (*mbd5/6* + MBD6^SWAP) plants and performed pollen RNA-seq and WGBS on multiple independent T2 lines. Since the effect of MBD6^SWAP, like *mbd5/6*, is likely only detectable in the VN of pollen, we expected the changes in whole pollen data to be diluted. Indeed, in both WT + MBD6^SWAP and *mbd5/6* + MBD6^SWAP, we observed a mild but consistent decrease in CG DNA methylation relative to the no-transgene control in the TSS region of genes upregulated in *mbd5/6* pollen, as well as over MBD6 ChIP-seq peaks (Fig. 6c, d, Supplementary Fig. 11e). This suggests that MBD6^SWAP can invade MBD5/6-bound regions and promote demethylation, though the demethylation phenotype is more mild than in *mbd5/6* mutants. Additionally, the effect of adding MBD6^SWAP was not stronger in the *mbd5/6* background compared to WT, suggesting that the presence of

native MBD5/6 doesn't inhibit MBD6^SWAP binding (Fig. 6c, d, Supplementary Fig. 11e). Over *mbd5/6* upregulated transcripts, the degree of demethylation was similar regardless of whether the locus could be transcriptionally rescued by loss of MBD7 ('rescued' vs. 'not rescued', Fig. 1d), confirming that MBD6^SWAP affects these loci equally regardless of whether they are normally MBD7 targets (Supplementary Fig. 11f). VN CG hypo DMRs with the highest MBD6 ChIP-seq signal were also slightly more demethylated on average in both WT + MBD6^SWAP and *mbd5/6* + MBD6^SWAP compared to no-transgene controls (Supplementary Fig. 11g), suggesting that MBD6^SWAP invades MBD5/6 targets proportionally to native MBD5/6 binding, and therefore accumulates more strongly at sites normally highly bound by MBD5/6. Overall, these data indicate that ectopically recruiting components of the MBD7 complex to MBD5/6-bound sites using a modified MBD6 is sufficient to cause partial demethylation of MBD5/6 target loci.

We next looked for expression changes in the RNA-seq data. We saw that transcripts upregulated in *mbd5/6* were also upregulated in WT + MBD6^SWAP pollen, but this effect was limited to the loci that are normally silent or barely expressed in wild-type (Fig. 6e, Supplementary Fig. 11h). Upregulation of lowly expressed *mbd5/6* upregulated loci also occurred regardless of whether the locus could be transcriptionally rescued by loss of MBD7 (Fig. 6e). Like the DNA methylation changes, the upregulation of these loci in WT + MBD6^SWAP was also substantially lower than in *mbd5/6* mutants. Additionally, loci that were already expressed in WT were unaffected in WT + MBD6^SWAP (Supplementary Fig. 11h). MBD6^SWAP also had no significant effect on expression when added to the *mbd5/6* background, where these loci are already upregulated (Fig. 6e, Supplementary Fig. 11h). Overall, these data show that MBD6^SWAP can recruit MBD7 complex components to MBD5/6 bound sites and promote partial demethylation and activation, but is less effective than loss of MBD5/6, and has little additional impact on transcription in *mbd5/6*.

### Discussion

In this manuscript, we identified an antagonistic interaction between the methyl-binding proteins MBD5 and MBD6, which maintain silencing in the VN of pollen, and the anti-silencer MBD7. This work sheds light on the mechanism of silencing by MBD5/6 and why it is uniquely observed in the VN of pollen, and also provides strong evidence that antagonism between MBD5/6 and MBD7 is only one of multiple interactions involved in mediating MBD5/6-based silencing. We also demonstrated the feasibility and power of performing simultaneous analysis of DNA methylation and RNA expression in single nuclei in plants. Developing pollen was an optimal system for testing the snmCT-seq protocol in plants, since vegetative nuclei undergo substantial changes to their methylome. Our snmCT-seq data was able to reveal the precise timing of these methylation changes relative to expression changes during VN maturation, and allowed us to clearly observe and characterize methylation changes in *mbd5/6* that occurred primarily in the VN and were hard to detect in bulk pollen. We foresee that this technology will help advance our understanding of

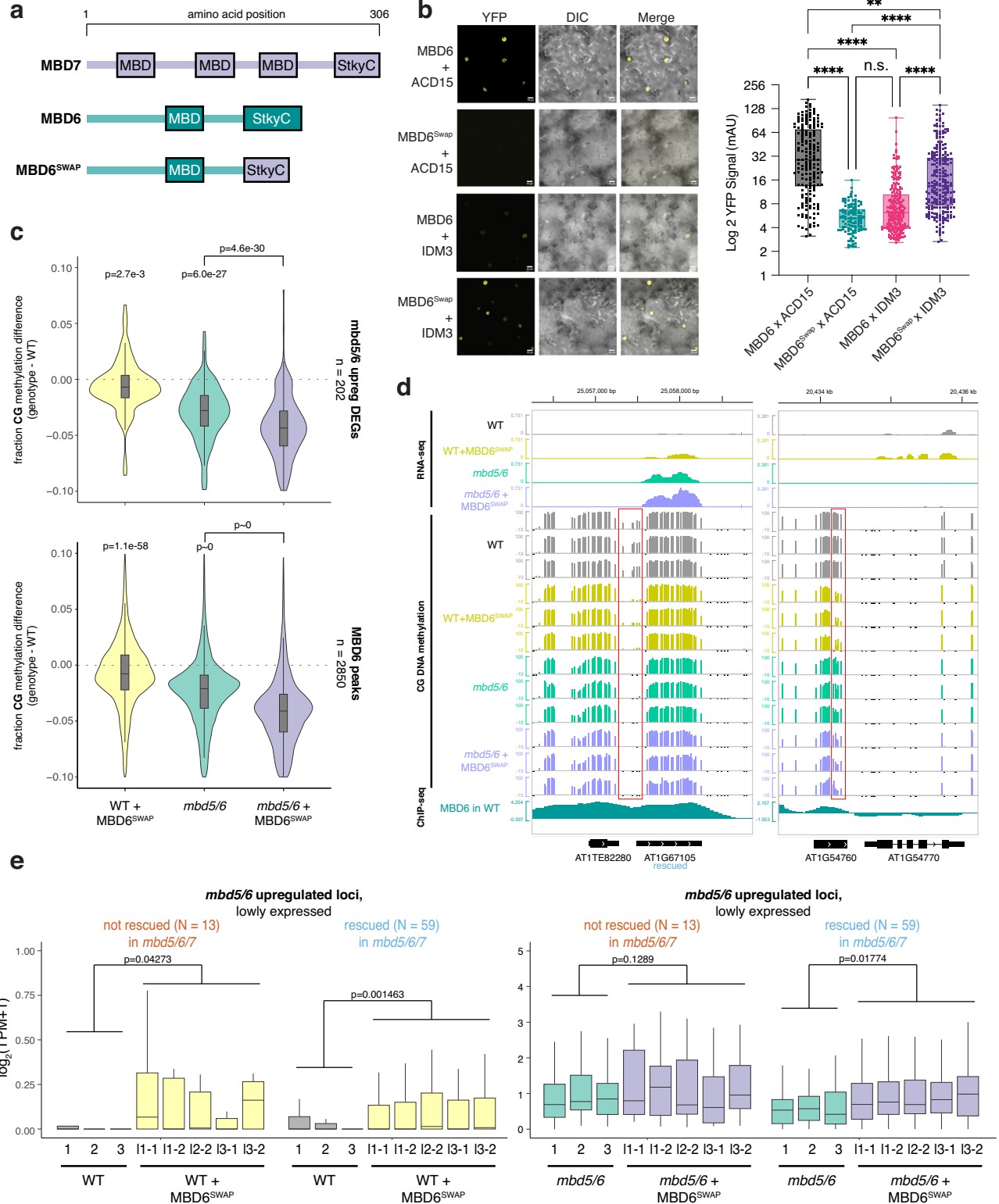

methylation dynamics in other heterogeneous plant tissues and potentially reveal other novel cell types that undergo methylation reprogramming.

While MBD5 and MBD6 were originally thought to act strictly downstream of DNA methylation[4], our work shows that a major function of MBD5/6 is to safeguard key loci like transposons from excessive demethylation by DME/ROS1 during normal VN reprogramming in pollen. Here we have focused on MBD7 as a key part of this mechanism. However, we note that MBD7 antagonism does not explain the entire

*mbd5/6* phenotype, since loss of MBD7 rescues only about half of the *mbd5/6* upregulated loci (Fig. 1d). Methylation changes in *mbd5/6* were rescued in *mbd5/6/7* at sites that are also transcriptionally rescued, but not consistently at other sites (Fig. 5b-d). This suggests that MBD5/6 protect their remaining targets from demethylation by either directly inhibiting DME and other demethylases, or interacting with other mechanisms that can promote DNA demethylation, similar to MBD7. Discovering how MBD5/6 does this will be an important focus for future work. Several other mechanisms involved in the epigenetic

**Fig. 6 | Recruiting the MBD7 complex to MBD5/6 targets causes mild deme-thylation and transcriptional upregulation of silent loci. a** Diagram of MBD6, MBD7, and MBD6[SWAP] constructs. **b** (left) Example BiFC images (YFP, DIC and DIC + YFP merge) and quantification of fluorescence (right) for indicated combi-nation of proteins. First protein listed was tagged with nYFP, second was tagged with cYFP. ****$p < 0.0001$, **$p < 0.01$, n.s. $p > 0.05$, Kruskal-Wallis one-way ANOVA with Dunn's multiple comparison test post hoc analysis. On right, each dot repre-sents a nucleus. Number of nuclei evaluated was: MBD6 x ACD15 $N = 187$ MBD6Swap x ACD15 $N = 117$ MBD6 x IDM3 $N = 229$ MBD6Swap x IDM3 $N = 235$. **c** Distribution of the difference in DNA methylation between wild-type (WT) and indicated genotype, average of three replicates per genotype. Values were calcu-lated over regions indicated at right of each plot. $p$-value = one-sided paired $t$-test.

**d** Example browser images of transcripts upregulated in WT + MBD6[SWAP] compared to WT, showing loss of methylation over regions occupied by MBD6. Left shows a locus upregulated in *mbd5/6* and rescued in *mbd5/6/7*, right is a locus not upre-gulated in *mbd5/6*. **e** Distribution of RNA-seq log$_2$(TPM) for all replicates from Col, Col + MBD6[SWAP], *mbd5/6*, and *mbd5/6* + MBD6[SWAP] (l1, l2, and l3 = independent transgenic lines). Values shown are for *mbd5/6* upregulated transcripts that had little to no expression in wild-type, separated based on whether they're rescued or not rescued in *mbd5/6/7* (Fig. 1d). $p$-value = two-sided paired $t$-test comparing the average of all samples within a genotype. **b**, **c**, **e** Boxplots report the median (central line), the 25th (Q1) and 75th (Q3) percentiles (box edges), and largest/smallest value within 1.5 * interquartile range (whiskers).

reprogramming of pollen vegetative cells have been described[11,14]. For example, a recent study found that another set of DNA methyl-binding proteins, SUVH4/5/6, interact with ARID1 to slow down the eviction of H3K9me2 from the VN, functioning in some ways similarly to MBD5/6 in preventing excessive DNA demethylation in VN[14].

Our data suggest that excessive loss of methylation in the *mbd5/6* VN can enable transcription of some loci, possibly by allowing methylation-sensitive transcription factors or transcriptional coacti-vator complexes to gain access to these methylated chromatin regions. This also likely explains why the *mbd5/6* derepression phe-notype has only been observed in the VN, since that is one of the few cell types where such dramatic methylation reprogramming is known to occur in Arabidopsis[7,12,13]. Linker histone H1 is also depleted in the VN[11]. H1 is involved in chromatin condensation, so its depletion in VN is thought to facilitate access of DME to some heterochromatic TE targets[11]. Consistent with this, both our single cell transcriptomic data (Supplementary Fig. 5f) and published imaging data[11] show that H1.1 and H1.2 expression levels decrease dramatically prior to pollen mitosis I and therefore are already depleted before DME and MBD5/6/7 are upregulated and before active demethylation of the VN begins. The FACT complex and its nucleosome remodeling activity are also required to allow access of DME to its heterochro-matic targets in endosperm[35]. Thus, in the majority of cell types MBD5/6's function in shielding DNA from demethylases could be functionally redundant with other factors that maintain tight chro-matin compaction and prevent access to the demethylation machinery. Indeed, we previously observed enhancement of the *mbd5/6* phenotype in *h1* mutant seedlings[6], suggesting that MBD5/6 become more important in a mutant where chromatin is less com-pacted and more accessible. Future experiments should test whether MBD5/6 are also required for silencing in other cell types that undergo active demethylation, like the central cell[36], and whether MBD5/6 also function by antagonizing MBD7 in these other tissues. Similarly, our data suggests that MBD7's activity is also prominent in the pollen VN due to chromatin decompaction. This reconciles well with the initial discovery of MBD7 as an antisilencer in genetic screens[15–18], because transgenes typically integrate in a euchromatic, open environment, and are targeted by the demethylation machin-ery, which works against de novo methyltransferases. Indeed, our MBD7 ChIP-seq data suggests that in wild-type, MBD7 is enriched in open chromatin compared to pericentromeres.

New targets and potential functions for MBD7 were also brought to light by looking at *mbd5/6/7* triple mutants. This revealed an inter-esting tug of war between MBD5/6 and MBD7 complexes, which work in opposite directions to inhibit or facilitate expression of shared tar-get loci, respectively. This seems to play a particularly important role in the VN, where DNA demethylation is facilitated. Our data suggest that MBD7 loss prevents some DNA demethylation from occurring in VN, particularly early in VN development (Fig. 5), suggesting a role for the MBD7 complex in helping to mediate normal VN reprogramming. However, in *mbd5/6/7* VN, DNA methylation levels either resemble wild-type or are hypomethylated as in *mbd5/6*, suggesting MBD7

functions largely downstream of MBD5/6 to regulate DNA methylation levels in VN. This is also consistent with the transcriptional data (Fig. 1d). Thus, our results position the MBD5/6 as a key inhibitor of DNA demethylation in pollen VN via inhibition of MBD7. However, we showed that MBD7's ChIP-seq enrichment does not increase upon depletion of MBD5/6 or SLN, suggesting that MBD5/6 do not directly affect MBD7 localization, so the exact interaction between MBD5/6 and MBD7 remains unclear. Since all ChIP-seq experiments were per-formed with inflorescence tissue, it is also possible that MBD7 locali-zation is affected by loss of *mbd5/6* specifically in the VN, which would be undetectable in data from whole inflorescences. Future technolo-gical advancements allowing high confidence detection of ChIP-seq enrichment in VN nuclei could unravel a different result. Alternatively, it is possible that MBD5/6 interfere with MBD7's activity instead of its recruitment to DNA, but the mechanism for this remains to be determined.

Why would a cell want to simultaneously target both an activator and a repressor to the same locus? Many of the loci upregulated in *mbd5/6* and rescued in *mbd5/6/7* are transposons that are lowly expressed in wild-type rather than silent (Fig. 1d, Supplementary Fig. 2). One proposed role for TE expression in the VN is to reinforce silencing in the sperm cells via the generation and movement of small RNAs from the VN to the SN[37–40]. The balance of repression by MBD5/6 and antisilencing by MBD7 together could help ensure low, stable TE expression for siRNA production, while preventing massive loss of silencing that could negatively impact the fitness of the VN. However, these factors alone are not essential for pollen development, as we did not observe any strong phenotype in any of the mutants. Since MBD5, MBD6 and MBD7 are expressed throughout the plant[6], we favor the hypothesis that MBD5/6/7 function in most plant cells. However, this does not preclude an important function in pollen. A number of redundant regulatory mechanisms have evolved to reinforce silencing of transposons and other dangerous elements, including MBD5/6, highlighting their importance.

Despite the differences in their functions, the MBD5/6 and MBD7 complexes share many structural similarities, suggesting an ancestral system that has evolved to carry out multiple different specialized functions. Both MBD5/6 and MBD7 directly interact with ACD proteins ACD15 and IDM3, respectively. In the MBD5/6 complex, ACD15 then recruits ACD21, which together mediate multimerization and accu-mulation of numerous copies of MBD5/6 complex members at methylated genomic regions[5]. Here we showed that IDM3 similarly regulates the accumulation of MBD7, suggesting that protein accu-mulation may be an important aspect of both complexes' function. However, MBD5/6 foci are completely abolished in *acd15* mutants, while MBD7 foci formation was only partially abolished in *idm3*. One possible explanation is that MBD7 retains enough methyl-DNA binding ability, due to its three MBD domains, to form foci even in the absence of IDM3, whereas MBD5/6 only have one MBD domain and therefore may be more affected by loss of multimerization through their ACD partners. In addition to the shared ability to mediate accumulation, our data suggest that ACD proteins confer functional specificity to the

different MBD complexes. Indeed, we show that altering MBD6 to recruit IDM3 instead of ACD15/21 is sufficient to turn this protein from a repressor to an activator.

## Methods

### Plant materials and growth conditions

All plants used in this study were in the Columbia-0 ecotype (Col-0) and were grown on soil in a greenhouse under long-day conditions (16 h light / 8 h dark). The following mutant lines were obtained from Arabidopsis Biological Resource Center (ABRC): *mbd7-3* (GABI_067_A09), *sln* (SALK_090484), *mbd6-1* (SALK_043927). The *mbd5/6* double mutant used in this study is the previously described *mbd5/6* CRISPR-1[4]. The *idm3/lil-1* and YJ control seeds were kindly donated by the Law lab[15].

### Generation of CRISPR/Cas9 mutants

The *mbd5 mbd6 mbd7* triple mutant was generated using the pYAO::hSpCas9 system[41] to edit *MBD5* and *MBD7* in the *mbd6-1* (SALK_043927) genetic background, as previously described in ref. 6. Briefly, we cloned sequentially a guide for the MBD5 gene (ACCGGA-GAACCCGGCTACTC) and a guide for the MBD7 gene (GCTATTCA-CAGACACTTGGC) into the pYAO::hSpCas9 plasmid by overlapping PCR. The plasmid was transformed into Agl0 agrobacteria, which were used to transform *mbd6-1* plants via floral dipping. Mutations were screened in T1 transgenic lines by Sanger sequencing. The plants used in this study are T2 or T3 Cas9 negative segregants, with homozygous mutations.

### Generation of transgenic lines

The FLAG-tagged transgenic lines used for ChIP-seq were generated by cloning the indicated genes including their endogenous promoter and introns into pENTR/D-TOPO vectors (Thermo Fisher) using an In-Fusion reaction (TaKaRa). For MBD7, the promoter region used was ~1 kb. The sequences were then transferred via a Gateway LR Clonase II reaction (Invitrogen, 11791020) into a pEG302-based binary destination vector including a C-terminal 3xFLAG epitope tag. Similarly, YFP and RFP constructs were generated by first cloning the desired gene + promoter into pENTR using In-Fusion, then performing a Gateway LR reaction into a t-DNA plasmid (pGW540) that adds a C-terminal YFP or RFP tag. MBD6$^{SWAP}$ transgenic plants were instead created by first generating [MBD6 endogenous promoter]-[MBD6 coding sequence lacking StkyC domain]-[MBD7 StkyC domain] in a pENTR/D-TOPO vector using In-Fusion (TaKaRa). This was inserted via a Gateway LR Clonase II reaction (Invitrogen, 11791020) into a t-DNA plasmid (pGW540) that adds a C-terminal YFP or RFP tag. For co-expression constructs (Supplementary Fig. 3f, 11c), the desired genes + promoters were first cloned upstream of YFP and RFP as described above. The RFP plasmid was then amplified using custom primers (cggaaccaattccc-gatctagtaaca and atcacaaaatctcttttgagtaacaaataaatt). This product was gel purified and inserted into the YFP plasmid that had been digested with XbaI.

Vectors were electroporated into Agl0 agrobacteria that were used for plant transformation by agrobacterium-mediated floral dipping. Successful FLAG line transformants were selected on soil using Basta resistance, while RFP/YFP lines were selected on 1/2x MS (Murashige and Skoog, Sigma) agar plates with hygromycin.

### RNA-sequencing

**Library preparation.** Mature pollen RNA-seq was carried out according to our previously described protocol[6]. We harvested ~500 µl of open flowers in 1.7 ml tubes (Eppendorf), then added 700 µl of Galbraith buffer (45 mM MgCl$_2$, 30 mM C$_6$H$_5$Na$_3$O$_7$.2H$_2$O [Trisodium citrate dihydrate], 20 mM MOPS, 0.1% [v/v] Triton X-100, pH 7, 5 µl per ml of 14.3 M Beta-mercaptoethanol [CAS-60-24-2, Fisher Scientific]), and vortexed the samples in the cold room for 2-3 min at max speed to

release the pollen from the anthers. The suspension was filtered with an 80 µm nylon mesh into a new 1.5 ml tube. The procedure was repeated one more time with the same flowers to increase the yield of pollen. The two aliquots of filtered pollen suspension were combined and centrifuged for 5 min at 500 *g*, 4 °C. The pollen pellet was frozen in liquid nitrogen and stored at −80 °C. This procedure was repeated on consecutive days to obtain replicates. Frozen inflorescence or pollen samples were disrupted with a tissue grinder and RNA extraction was performed with the Zymo Direct-zol RNA MiniPrep kit (Zymo Research). In both cases, the in-column DNase I digestion was performed. RNA-seq libraries were generated using the TruSeq Stranded mRNA Library Prep Kit (Illumina), following the manufacturer's instructions and starting with 500–1000 ng of RNA as input, and sequenced on a Novaseq 6000 machine.

**Analysis.** RNA-seq reads were filtered based on quality score and trimmed to remove Illumina adapters using TrimGalore[42] v0.6.7. Reads were then aligned to the Arabidopsis reference genome (TAIR10) using STAR[43] v.2.7.9a using custom annotations generated previously based on pollen RNA-seq data[6] (https://github.com/clp90/mbd56_pollen/), and allowing up to 5% of mismatches (-outFilterMismatchNoverReadLmax 0.05). Only uniquely aligned reads were used. PCR duplicates were removed using MarkDuplicates (v1.121) from the Picard Tools suite[44]. Coverage tracks for visualization in the genome browser were generated using Deeptools[45] version 3.0.2 bamCoverage with the options --normalizeUsing CPM and --binSize 10. HTseq[46] was used to obtain read counts over genes and transposons in the custom pollen annotation, using the option --mode=intersection-strict. Differential gene expression analysis was performed using DEseq2[47] v1.34.0 with a cutoff for significance of adjusted *p*-value < 0.05 and |log2FC| > 1. All transcript types were analyzed together in the differential expression analysis (genes, TEs, and other undefined non-coding transcripts). Transcripts per million (TPM) values were estimated using StringTie version 2.1.6[48]. Figures were generated using R version 4.4.0 and the packages *ggplot2* version 3.5.1[49] and *pheatmap* version 1.0.12[50].

### Bisulfite sequencing

Mature pollen samples for bisulfite sequencing were obtained with the same procedure described for RNA-seq. DNA extraction from pollen pellets was either performed with the cetyl trimethylammonium bromide (CTAB) method and treated with RNase A (Qiagen) (batches lib1-lib4, Supplementary Data 1) or using the Qiagen DNeasy Plant mini kit with on-column RNase A digestion (batch lib5, Supplementary Data 1). 20–100 ng of DNA per sample were sheared with the Covaris S2 instrument to an average fragment size of 200 bp (Duty Cycle = 10%, Intensity = 5, Cycles per Burst = 200, Treatment Time = 120 s). Sheared DNA was used as input for library preparation using the Nugen Ultra-low Methyl-Seq kit, following manufacturer's instructions except for the bisulfite conversion, which was done with the EpiTect Bisulfite Kit (Qiagen, 59104). The bisulfite conversion reaction in the thermocycler was performed twice (overnight) to increase efficiency. Final libraries were sequenced on NovaSeq 6000 or NovaSeq X plus.

**Analysis.** Reads were trimmed and filtered with TrimGalore[42] and mapped to the TAIR10 genome with Bismark[51] v2.1.6. PCR duplicates were removed using the script provided with Bismark. Bismark was also used to obtain the methylation percentages for each cytosine and to generate the per-position DNA methylation tracks. For downstream analyses, only cytosines with at least 5 aligned reads were used. Average methylation (weighted by coverage) over different sets of genomic regions was obtained using a custom script (sumByFeature.sh, provided in linked github repository). Plots were generated using R version 4.4.0 and the package *ggplot2* version 3.5.1[49]. Published fastq files

from buds whole-genome bisulfite sequencing (SRR6410887, SRR6410888)[25] were downloaded from GEO and reanalyzed as above. Per-position methylation data for both replicates was then pooled by adding together the methylated/unmethylated read counts for both replicates at each position.

## Chromatin immunoprecipitation sequencing (ChIP-seq)

ChIP-seq was performed with about 2 grams of inflorescences with unopened flower buds per sample, which was collected and flash-frozen in liquid nitrogen until use. ChIP was performed as in ref. 52. All buffers, unless otherwise indicated, were supplemented with protease inhibitors: PMSF (Sigma), benzamidine (Sigma), cOm-pleteTM Protease Inhibitor Cocktail (Sigma), and MG132 (Sigma). Tissue was ground using a TissueLyser II (Qiagen), using 2 rounds of 90 seconds at frequency 1/30. Tissue powder was dissolved in 25 mL nuclei isolation buffer (50 mM HEPES, 1 M sucrose, 5 mM KCl, 5 mM MgCl$_2$, 0.6% Triton X-100). Formaldehyde was added to dissolved sample to a final conc. of 1% and incubated 12 min at room temp on a rotator mixer. Crosslinking was quenched with freshly prepared 2 M glycine for 5 min room temp on a rotator mixer. Nuclei were then purified by first filtering through one layer of miracloth, then successively pelleting and resuspending nuclei in extraction buffer 2 (0.25 M sucrose, 10 mM Tris-HCl pH 8, 10 mM MgCl2, 1% Triton X-100, 5 mM βME) and extraction buffer 3 (1.7 M sucrose, 10 mM Tris-HCl pH 8, 2 mM MgCl$_2$, 0.15% Triton X-100, 5 mM βME). After resuspending in extraction buffer 3, nuclei were pelleted for 1 h at 12,000 $g$ 4 °C, then resuspended in nuclei lysis buffer (50 mM Tris pH 8, 10 mM EDTA, 1% SDS). Chromatin was sheared using a Bioruptor Plus (Diagenode) sonicator, 30 s on / 30 s off on high, 22 cycles. Samples were diluted with ChIP dilution buffer (1.1% Triton X-100, 1.2 mM EDTA, 16.7 mM Tris pH 8, 167 mM NaCl), and incubated with anti-FLAG antibody (Monoclonal ANTI-FLAG M2, cat. no. F1804, Sigma, 1:800 dilution) overnight at 4 °C on a rotator mixer. Samples were then incubated with a 1:1 mixture of Protein A and Protein G Dynabeads (Invitrogen) for 2 h at 4 °C on a rotator mixer. Beads were then washed twice with low salt buffer (150 mM NaCl, 0.2% SDS, 0.5% Triton X-100, 2 mM EDTA, 20 mM Tris pH 8), once with high salt buffer (200 mM NaCl, 0.2% SDS, 0.5% Triton X-100, 2 mM EDTA, 20 mM Tris pH 8), once with LiCl buffer (250 mM LiCl, 1% Igepal, 1% sodium deoxycholate, 1 mM EDTA, 10 mM Tris pH 8), and once with TE buffer (10 mM Tris pH 8, 1 mM EDTA). Sheared chromatin was eluted from beads by resuspending in 250 uL elution buffer (1% SDS, 10 mM EDTA, 0.1 M NaHCO$_3$) and incubating at 65 °C for 20 min at 1000 rpm. This step was performed twice to increase yield, combining eluates (final volume 500 uL). 20 uL of elution was used for Western Blot to check pull-down using anti-FLAG HRP antibody (1:10,000 dilution, Sigma, Monoclonal ANTI-FLAG M2-Peroxidase, cat. no. A8592). Reverse crosslinking was performed by adding 20 uL 5 M NaCl and incubating 65 °C overnight. Protein was digested by adding 1 uL Proteinase K (Invitrogen), 10 uL 0.5 M EDTA and 20 uL 1 M Tris pH 6.5 and incubating 4 h at 45 °C 400 rpm. DNA was purified using a standard phenol:chloroform:isoamyl alcohol extraction and precipitated with sodium acetate at ethanol overnight at −20 °C. Libraries were made using the Ovation Ultra Low System V2 1-16 kit (NuGEN, 0344NB-A01) following the manufacturer's instructions, with 15 cycles of PCR, and sequenced on NovaSeq X Plus.

**Analysis.** Reads were trimmed and filtered with TrimGalore[42] and mapped to the TAIR10 genome with bowtie2[53] v2.3.4.3 with parameter -X 1000. For each sample, genome-wide coverage tracks were generated using Deeptools[45] v3.5.5 bamCoverage with options −binSize 10 −normalizeUsing CPM −extendReads and excluding the mitochondrial and chloroplast genomes. Coverage tracks for the two no-FLAG controls were averaged using Deeptools

bigwigAverage, and then each sample was normalized to the averaged control track using Deeptools bigwigCompare with −operation log2 and −binSize 10. Peaks were identified using MACS2[54] v3.0.0a7 callpeak with options -g 119000000 --seed 123456 --broad -f BAMPE --broad-cutoff 0.1. Peaks from replicates were combined using bedtools merge. ChIP-seq signal over 400 bp bins was calculated using Deeptools multiBigwigSummary, while metaplots of ChIP-seq signal over different genomic regions was performed using Deeptools computeMatrix followed by plotHeatmap. Clustering of peaks was performed using the −kmeans option of plotHeatmap with k = 5. Published raw sequencing data were downloaded from GEO and reanalyzed as described above: MBD5-Myc and MBD6-Myc (GSM5267036, GSM5267037, GSM5267038)[4], H3K9me2 (GSM3130575, GSM3130576)[55], Pol V (GSM2667838, GSM2667837)[56].

*mC density correlations:* For mC density correlations with ChIP-seq signal (Supplementary Fig. 3c), the genome was split into 400 bp non-overlapping bins tiled genome-wide. For analysis shown in Fig. 2a, methylation density was instead calculated over 400 bp bins centered around the midpoint of the ChIP-seq peak. For each bin, methylation density in each sequence context (CG, CHG and CHH) was calculated as the sum of the methylation percentage of all cytosines in that context in the bin. Methylation data used from Harris et al. 2018 Science (SRR6410887, SRR6410888)[25]. Plots were generated in R using the geom_smooth() function from the ggplot2[49] package with default parameters.

## snmCT-seq

Sample preparation for snmCT-seq was performed as previously described for snRNA-seq with the 10X genomics platform, with a minor variation in the last step prior to nuclei sorting[6]. Briefly, mixed spores were isolated from open flowers, and pollen was broken to release nuclei into solution. Nuclei were then filtered and collected by centrifugation. Nuclei pellets were resuspended in 2 mL nuclei extraction buffer and DAPI buffer combined in a 1:8 ratio (Cystain UV Precise P, Sysmex). The Col and *mbd5/6* samples were then split into two aliquots, which were sorted simultaneously on two machines: a BD FACS ARIAII instrument and BD FACS ARIA, using the 70 μm nozzle. For each experiment, 4 plates were sorted per genotype, then samples were swapped to the other machine to sort an additional 4 plates. The library preparation was performed as previously described with specific modifications described below[27]. Single nuclei were sorted into a 1 μl reverse transcription reaction that contains 1X First-Strand buffer (Invitrogen) supplemented with 2.5 mM MgCl$_2$, 5 mM DTT, 0.1% Triton X-100, 0.38 mM each dATP, dTTP, dGTP and 5-methyl-dCTP (NEB), 1 μM dT30VN_5 (/5Biosg/AAGCAGUGGUAUCAACGCAGAGUA-CUTTTTTTUTTTTTTUTTTTTTUTTTTTTTVN, synthesized and HPLC purified by IDT), 2 μM N6_3 (/5Biosg/AAGCAGUGGUAUCAACGCAGA-GUACNNNNNN, synthesized and HPLC purified by IDT), 2 μM TSO_4: (/5Biosg/AAGCAGUGGUAUCAACGCAGAGUGAAUrGrGrG, synthesized and HPLC purified by IDT), 0.4 U RNaseOUT (Invitrogen), 0.2 U SuperaseIn (Invitrogen), 2 U Superscript II (Invitrogen). The reverse transcription reaction was incubated in a thermocycler with the following condition: 25 °C for 5 min, 42 °C for 60 min, 70 °C for 15 min followed by 4 °C incubation. The cDNA was amplified by adding 3 μl PCR mix into each reverse transcription reaction. The PCR mix contains 0.8 μl 5X KAPA 2 G Robust Buffer A, 0.2 μl 12 μM ISPCR23_3 (/5SpC3/AAGCAGUGGUAUCAACGCAGAGU), 0.016 μl 5U/μl KAPA2G Robust HotStart DNA Polymerase. The PCR reaction was performed using the following cycling condition: 95 °C for 3 min for the initiation denaturing, 8 cycles amplification (95 °C for 15 s, 60 °C for 30 s, 72 °C for 120 s), followed by a final elongation at 72 °C for 5 min. Excessive DNA oligos were digested using an 1 μl mix of 0.5 μl of 2 U/μl Uracil DNA Glycosylase and 0.5 μl of EB buffer (Qiagen cat. #19086) at 37 °C for 30 min. The protocol of snmCT-seq is available on protocol.io at

https://www.protocols.io/view/snmcat-v2-x54v9jby1g3e/v2/. The snmCT-seq library was amplified for 20 cycles to account for the smaller genome size of Arabidopsis (130 Mb).

## snmCT-seq data analysis

**Initial alignment, quality filtering, and data extraction.** Each library consists of a pool of 384 individual libraries from a single 384 well plate, containing single sorted pollen nucleus in each well. Sixteen plates were sorted each for Col and *mbd5/6*, across two separate experiments, for a total of 32 plates/pools of 384 single nuclei each. Note that the library prep for plate #8 of expt 2 Col, and plate #1 of expt 2 *mbd5/6* failed, and so these plates were not analyzed. Final pools were each sequenced using a paired-end 100 bp x 100 bp protocol on Novaseq 6000, obtaining an average of approximately 70 M reads with valid well barcodes per pool in expt 1, and 124 M in expt 2 (Supplementary Data 1).

Each pool was initially processed using a custom pipeline (snmCT_seq_map.sh) which handled well demultiplexing, quality filtering, alignment, and extraction of WGBS per-position methylation data and RNA-seq per-gene count data. Briefly, read pairs were demultiplexed according to well barcode, with the same list of barcodes used for each plate (snmCAT_well_barcodes.txt). As recommended in Luo et al. 2017[57], paired-end reads were hard trimmed by 10 bp at both 5′ and 3′ ends to remove bias due to random priming and adaptase behavior, and the 5′ end of read 1 was trimmed by an additional 8 bp to remove the well barcodes. Additionally, due to high rates of chimeric transcripts, read pairs were treated as separate reads. Demultiplexed reads were then trimmed for any remaining adapter sequences and filtered to remove low quality sequences using trim_galore[42] with options `-a AGATCGGAAGAG -q 25 --stringency 3`. Remaining reads were aligned to the transcriptome using STAR[43] with options `--alignEndsType EndToEnd --outFilterType BySJout --outFilterMultimapNmax 1 --winAnchorMultimapNmax 1 --outFilterMismatchNmax 999 --outFilterMismatchNover ReadLmax 0.05 --alignIntronMin 70 --alignIntronMax 5000 --outFilterIntronMotifs RemoveNoncanonical`, retaining only unique alignments. Reads that did not align to the transcriptome were then aligned to the bisulfite-converted genome using Bismark[51] with options `-N 0 -L 20 --non_directional`, again retaining only unique alignments. All RNA-seq and WGBS alignments were filtered to remove PCR duplicates using the MarkDuplicates function from the Picard tools suite[44]. RNA-seq read counts over genes were obtained using htseq-count[46], and WGBS data was processed using the Bismark methylation extractor function to obtain per-site methylation data. Wells/nuclei were then filtered using a very basic, permissive filter to remove failed wells. This filtering retained all wells with at least 1000 RNA-seq reads over at least 200 genes and at least 10% of 1 kb bins tiled genome-wide with 1 + WGBS read. All steps above were handled by our custom pipeline (snmCT_seq_map.sh, see code availability section).

**snmCT-seq transcriptome-based analysis and clustering.** Count data from all wells passing the basic RNA QC listed above were combined into one count matrix per sample (4 total). We then applied an additional filter designed to detect potential doublets (two or more nuclei stuck together and sorted into a single well). For this filter, we used the WGBS data for each nucleus to count the number of sites in the genome covered by at least one read, exactly one read, and more than one read. Since most nuclei in the dataset should be haploid (except GN post S-phase), and PCR duplicates have been removed, >1 coverage at any site should not occur. Indeed, coverage >1x was rare, but increased with total genome coverage (Supplementary Fig. 12a). 163 nuclei across the dataset with substantially elevated rates of >1x coverage were censored as likely doublets (Supplementary Fig. 12a, red dots, Supplementary Data 3). Count data from remaining nuclei were then loaded into Seurat 4.0.4[28] and processed following a similar approach to the "Analysis of snRNA-seq" section in the methods of Ichino et al. 2022[6]. Briefly, nuclei were first filtered to exclude those with elevated rates of reads coming from the chloroplast and mitochondrial genomes ( >10% of RNA-seq reads) (Supplementary Data 3). We then ran the standard Seurat analysis pipeline to perform initial scaling and finding variable features, then integrated the four samples using the IntegrateData() function after selecting integration features and finding integration anchors. We then performed UMAP projection using RunUMAP() with reduction = 'pca', dims = 1:20, min.dist = 0.5, and performed clustering using FindNeighbors() and FindClusters() with resolution = 1.5 to obtain an initial clustering (Supplementary Fig. 12b). We combined the three SN and the three GN clusters into single clusters due to similar marker expression. The "MN to VN" cluster was named based on having intermediate expression of both MN and VN1 markers, and the"VN1 to 2" cluster was named based on having intermediate expression of both VN1 and VN2 markers. This preliminary clustering also included a cluster that we labeled "VN and SN", for having expression of both mature VN (VN4 + VN5) and SN markers (Supplementary Fig. 12c). Upon closer inspection, it became clear that mature VN + SN doublets were common in the dataset and were not all detected in our prior coverage-based filtering (Supplementary Fig. 12a). This is likely because, in mature pollen, sperm nuclei often form physical associations with vegetative nuclei[58], and so may be more likely to be sorted together. As a result, a substantial fraction of nuclei in the mature VN and SN clusters had relatively strong expression of markers for both clusters, as did nuclei in the "VN and SN" cluster (Supplementary Fig. 12c). Additionally, average CG DNA methylation over certain regions is much higher in sperm than in VN[13], but our initial clustering revealed a subpopulation of SN with lower CG methylation, and a subpopulation of VN4 and VN5 with elevated CG methylation (Supplementary Fig. 12d). We therefore applied an additional filter by calculating the $\log_2$ ratio of average expression of SN markers, over average expression of VN4,5 markers, for each nucleus in VN4, VN5, and SN (Supplementary Fig. 12c). All nuclei with a $\log_2$ ratio within [−0.2,0.2] (Supplementary Fig. 12c, red dotted lines), indicating similar levels of mature VN and SN marker expression, were reassigned to the "VN and SN" cluster and censored from most analyses.

**Analysis of third replicate of snmCT-seq.** Initial alignment, quality filtering, and data extraction was performed as above. Preliminary doublet filtering based on WGBS coverage was also performed as above (Supplementary Fig. 12e). Count data from remaining nuclei were then loaded into Seurat 4.0.4[28], and preliminary filtering to exclude wells with high chloroplast or mitochondrial content was performed as for expts 1 and 2 above (Supplementary Data 3). To obtain cluster assignments, the RNA counts matrix from each genotype in expt 3 was then projected onto the expt 1 + 2 UMAP using the Seurat function FindTransferAnchors() with parameters `dims = 1:20, reference.reduction = "pca"`, and using only features present in both datasets, followed by MapQuery() with parameters `reference.reduction = "pca", reduction.model = "umap"`. Additional doublet filtering based on expression of mature VN and SN markers was also carried out on the nuclei assigned to mature VN or SN, as above (Supplementary Fig. 12f).

**snmCT-seq DNA methylome data analysis.** Prior to snmCT-seq DNA methylome analysis, we performed additional filtering to remove nuclei ($n = 160$ across expts 1 and 2, $n = 124$ for expt 3) with likely poor bisulfite conversion rates, based on elevated chloroplast methylation (Supplementary Fig. 13a–d). After finalizing the clusters based on the RNA-seq data, the per-site methylation data for all nuclei in each cluster was pooled to obtain methylation tracks for each cluster using a custom script (see code availability section below). Methylation metaplots over different sets of peaks/genes/regions were generated

with deeptools plotProfile[45] and chromosome-wide plots were made as described for bulk BS-seq. Additionally, average methylation per-nucleus over specific regions (e.g. VN CG hypomethylated DMRs) was calculated as the weighted mean (sum methylated / (sum methylated + sum unmethylated)) across all covered cytosines within the regions using a custom script (see code availability section below).

For unsupervised clustering based on single-nucleus methylomes alone, we generated.allc files from the initial bismark BAM alignment files using the bam-to-allc function of ALLCools[59]. The fraction of methylated cytosines in each context (CGN, CHG, CHH) was calculated over 25 kb bins tiled genome-wide, by dividing the number of methylated cytosines in the correct context across all reads in the bin by the number of total cytosines in the correct context detected in the bin. UMAP dimensionality reduction was performed using Scanpy[60].

### Gene and TE homology analysis for *mbd5/6* upregulated loci (Supplementary Fig. 2)

Each of the 202 *mbd5/6* upregulated loci (Fig. 1d) were blasted separately to the Araport11[61] gene, TE, and TE gene annotations using blastn[62] and output in format 6. For each locus, if there was no significant homology to a gene, and the TE or TE gene bitscore was > 100, the locus was classified as having TE homology. If there was no significant TE or TE gene homology, but a gene bitscore > 100, the locus was classified as having gene homology. For loci with some homology to both genes and TE/TEgenes, if one bitscore was at least 3x higher than the other, the locus was classified as belonging to the higher category. Six loci had no significant TE or gene homology and were classified as 'neither'.

### Statistics and reproducibility

For RNA-seq, at least three replicates were performed per condition. For WGBS, each condition was assayed at least twice, except for *lil* which was only done once. For ChIP-seq, each genetic background was assayed at least twice using independent transformed lines. A few ChIP-seq samples were excluded from the analysis due to poor expression of the tagged protein or poor capture during ChIP-seq (see Supplementary Fig. 3a). For snmCT-seq, nuclei were sorted into between six and eight 384 well plates per sample per experiment. The experiment was repeated three times for Col and *mbd5/6*, and once for *mbd7* and *mbd5/6/7*. Poor quality nuclei were censored based on a number of filtering parameters (see *snmCT-seq data analysis* above). No statistical method was used to predetermine sample size. Most analyses were performed in R. For some analyses with high N (100 s of nuclei or 1000 s of genomic regions), effect size (measured as Cohen's D) was used to evaluate the impact of the tested condition alongside or in lieu of running a statistical test to estimate *p*-value. Specific tests are indicated in the legends of each figure. For imaging data, all underlying data shown in figures (foci counts, intensities etc.) are provided in Supplementary Data 5.

### Generating diagrams

BioRender was used to generate part of the diagram in Figs. 3a, d, and Supp. Fig. 7a (see figure legends). All remaining diagrams were generated in Adobe Illustrator.

### Reporting summary

Further information on research design is available in the Nature Portfolio Reporting Summary linked to this article.

## Data availability

The sequencing data generated in this study have been deposited in the Gene Expression Omnibus (GEO) database under accession code GSE275832. Published data used in this study: MBD5-Myc and MBD6-Myc (GSM5267036 [https://www.ncbi.nlm.nih.gov/geo/query/acc.cgi?acc=GSM5267036], GSM5267037, GSM5267038)[4], H3K9me2 (GSM3130575 [https://www.ncbi.nlm.nih.gov/geo/query/acc.cgi?acc=GSM3130575], GSM3130576)[55], NRPE1 (GSM2667838 [https://www.ncbi.nlm.nih.gov/geo/query/acc.cgi?acc=GSM2667838], GSM2667837)[56], 10x snRNA-seq of pollen nuclei (GSE202422 [https://www.ncbi.nlm.nih.gov/geo/query/acc.cgi?acc=GSE202422])[6]. The raw imaging data generated in this study are available in the Dryad database [https://doi.org/10.5061/dryad.vx0k6dk23]. Seurat objects for the snmCT-seq data analysis are available on GitHub at: https://github.com/clp90/MBD_antagonism. All additional data used to generate figures are provided in Supplementary Data 1-5.

## Code availability

All custom scripts used for this project are available from GitHub at: https://github.com/clp90/MBD_antagonism.

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

## Acknowledgements

We thank S. Feng, M. Akhavan, and the UCLA Broad Stem Cell Research Center BioSequencing core for high throughput sequencing, and Felicia Codrea and the Eli and Edythe Broad Center of Regenerative Medicine and Stem Cell Research UCLA Flow Cytometry Core Resource for nuclei sorting. We also thank Yan He and other members of the Jacobsen lab for assistance with ChIP-seq and helpful discussion, and C. Liu and M. Heffel for help with data analysis. This work was supported by a Ruth L. Kirschstein National Institutes of Health Award (F32GM136115) to C.L.P., a Philip Whitcome Pre-Doctoral Fellowship in Molecular Biology to L.I., an NSF GRFP (DGE-2034835, DGE-2444110) to T.J.B., and work in the Luo lab was supported by the UCLA DGSOM startup fund. S.E.J. is an Investigator of the Howard Hughes Medical Institute.

## Author contributions

L.I. and S.E.J. initially conceived the project. L.I., C.L.P., and S.E.J. wrote the manuscript. All authors participated in manuscript editing. L.I. performed most of the experiments with assistance from J.Y. snmCT-seq experiments were carried out by L.I., C.L.P., T.B., K.D.A., and Y.Z. ChIP-seq experiments were performed by C.L.P. and T.B. with assistance from S.M.L.S. MBD6[SWAP] experiments were conceived by BAB and carried out by B.A.B., T.B., N.J.B., and SMLS. Data analysis was performed mostly by C.L.P. with help from L.I., B.A.B., and K.D.A. Work was supervised by C.L. and S.E.J.

## Competing interests

The authors declare no competing interests.
