## [Transparent Peer Review file · Nature Communications]

An antagonistic epigenetic mechanism regulating gene expression in pollen revealed through single-nucleus multiomics

Corresponding Author: Dr Steven Jacobsen

Version 0:

Reviewer comments:

Reviewer #1

(Remarks to the Author)

In this work, the authors studied the opposing roles of Arabidopsis MBD5/MBD6 and MBD7 in regulating DNA methylation and transcription. They found that loss of MBD7 rescues transcriptional defects at a subset of loci bound by MBD5/6. Single-nucleus profiling revealed that MBD5/6-bound loci undergo active demethylation in early VN, which is accelerated in *mbd5/6* mutants before transcriptional derepression occurs. In the absence of MBD5/6, MBD7 binds ectopically, correlating with further demethylation and transcriptional activation, which is reversed when MBD7 is lost. Artificial recruitment of MBD7 to MBD5/6 targets similarly induces partial demethylation and transcriptional activation. This study proposes that MBD5/6 maintain silencing by preventing MBD7-mediated enhancement of active demethylation during VN development. MBD proteins are critical regulators of DNA methylation and transcription, with MBD5/MBD6 and MBD7 known to bind highly methylated genomic regions. However, their functional interactions remain poorly understood. This study presents a compelling model of antagonistic binding between MBD5/MBD6 and MBD7, predominantly in heterochromatic regions, to modulate local DNA methylation and transcription. This research was elegantly designed, featuring well-performed experiments with appropriate controls, thorough data analysis, and conclusions strongly supported by the results. While there are no major concerns, an intriguing question arises regarding H1 expression dynamics during pollen development. It would be insightful to determine the exact timing of H1 absence in early VN, coinciding with DNA methylation changes and the activity of MBD5/6/7, according to their snmRNA-seq data.

Reviewer #2

(Remarks to the Author)

It is well established that during pollen development, the vegetative nucleus (VN) undergoes significant nuclear condensation and epigenetic reprogramming. Previous studies have identified various mechanisms involved in regulating these processes, but their complexity warrants further in-depth investigation. Methyl-CpG-binding domain (MBD) proteins are known to recognize DNA methylation sites and potentially influence subsequent transcriptional activity. Among these, MBD5 and MBD6, which are mildly expressed in the VN, have been implicated in VN function, although their exact mechanisms remain unclear.

Using simultaneous single-nucleus methylcytosine and transcriptome analysis, Picard et al. demonstrated that MBD5 and MBD6 actively repress transcription through DNA methylation in the early VN following the first pollen mitosis. In contrast, their nearly homologous protein, MBD7, counterbalances this repression by promoting DNA demethylation, thereby preventing excessive transcriptional silencing. Notably, the absence of MBD7 can rescue the activation of transposable elements (TEs) in *mbd5/6* mutants. MBD5 and MBD6 maintain transposon silencing in the VN by inhibiting the MBD7 complex from amplifying active demethylation during VN maturation.

These findings further support the concept that the VN avoids excessive de-repression through finely tuned regulatory mechanisms, such as the MBD5/6-mediated silencing pathway.

Major concerns:

1. Although exploring how epigenetic reprogramming is regulated during male gametophyte development is an intriguing perspective, the biological significance of MBD5/7-mediated silencing in the VN remains unclear.
2. In Figure 1a, of the 202 transcripts upregulated in the *mbd5/6* mutant pollen, how many correspond to TE loci and how

many to gene loci? Do the *mbd56* and *mbd567* mutants exhibit any pollen-related phenotypic abnormalities or alterations in VN nuclear condensation? Considering the subsequent text mentions that *mbd56*-associated methylation and transcriptional dysregulation primarily occur in the VN, do the *mbd56* and *mbd567* mutants display any abnormalities in pollen tube germination?

3. In Fig. 1d and Fig. 2a-d, the authors concluded that MBD7 redistributes to MBD5/6 binding loci, particularly localizing to the pericentromeric regions with higher H3K9me2 levels. Considering that H3K9me2 persists in the early VN before its subsequent depletion, as demonstrated by a recent study (Li et al., 2004; 10.1038/s41467-024-51513-4), were there any changes in H3K9me2 levels in *mbd7*, *mbd5/6*, and *mbd5/6/7* mutants in the early post-mitotic VN?

4. In Figure 2a, the authors present the signal intensity across all peaks bound by MBD6 and MBD7. To further support the antagonistic roles of these proteins and their functions in pericentromeric heterochromatin, the authors should consider including a heatmap or metaplot to visualize the signal intensity and distribution of MBD6 on MBD7 peaks and vice versa. Additionally, categorizing MBD6 and MBD7 peaks into euchromatic regions and pericentromeric heterochromatic regions could help elucidate changes in binding intensity between the two proteins at their reciprocal peaks. Such analyses would provide deeper insights into the distinct and overlapping functions of MBD6 and MBD7.

5. In addition to the MBD6 and MBD7 signals in roots presented in Fig. 2 and Extended Data Fig. 3, have the authors examined their signals and colocalization in the VN? Furthermore, have any changes in the signal levels or foci of MBD7 been observed in *mbd56* pollen, despite its transcription levels showing no significant variation?

6. In Fig. 3 and Fig. 4, the authors demonstrated that *mbd5/6* mutants lose additional methylation in the early post-mitotic VN prior to gene upregulation. What is the function of these upregulated genes that require silencing in the early VN, especially considering that their methylation is later lost due to DME activity? Furthermore, MBD5/6 targets were preferentially localized to the highly methylated pericentromeric regions enriched with transposons. Could the premature activation of these transposons in the VN negatively impact VN function or GC development?

7. Many RNA sequencing analyses, localization experiments, and Co-IP assays should be conducted using pollen rather than inflorescences or roots, as the authors themselves noted that MBD56-mediated silencing exhibits strong cell-type specificity.

8. In Extended Data 6A, CHG methylation in the SN is shown to be higher than in the VN. However, according to Walker et al. (2018; 10.1038/s41588-017-0008-5), Calarco et al. (2012; 10.1016/j.cell.2012.09.001), and Ibarra et al. (2012; 10.1126/science.1224839), who performed methylation sequencing based on FACS, total CHG methylation in the VN is reported to be higher than in the SN. Could the authors please explain this discrepancy?

9. The authors should consider including snmCT-seq results from the VN of the *mbd567* mutant and compare the methylation levels in the VN between *mbd56* and *mbd567* mutants, rather than focusing solely on the methylation levels in mature pollen of the *mbd567* mutant.

10. Are the amino acids in the StkyC domain of MBD6 and MBD7 conserved?

11. Was there any phenotypic change or disruption in developmental pathways when the VN exhibited faster depletion of the CG methylome in the *mbd5/6* background?

12. Based on the observation that not only MBD56, but also heterochromatin factors SUVH4/5/6 and their interacting protein ARID1, which are widely expressed in the VN as shown by a recent study (Li et al., 2024; 10.1038/s41467-024-51513-4), the authors may consider discussing the possible biological significance of these findings.

13. Two papers on small RNA movement in pollen should be cited: Wu et al., 2021 (doi: 10.1111/nph.16871) and Herridge et al., 2024 (doi: 10.1038/s41594-024-01392-6).

Reviewer #3

(Remarks to the Author)

Reviewer summary:

In this manuscript, the authors describe an antagonistic relationship between three genes that regulate DNA methylation in pollen. They explored an epigenomic dataset generated for the double mutant, *mbd5/6*, a triple mutant, *mbd5/6/7* and wild-type and found that a large number of loci that are de-methylated and up-regulated in the double mutant were “rescued” by the loss of MBD7. They supported their assertion that MBD7 de-methylates genes that are bound (and kept methylated by) MBD5/6 by using ChIP-seq profiling of MBD6 and MBD7 in WT and mutant lines. They went on to profile different pollen nuclei cell states using the recently-developed snmCT-seq which profiles both cytosine methylation and gene expression of single nuclei. Using this data, they piece together a developmental trajectory and found that demethylation is somewhat specific to the Vegetative Nuclei (VN) and progresses according to developmental time. Lastly, the authors swapped the complex recruitment domain of MBD6 and MBD7, showing that the StkyC domain of MBD7 can ectopically demethylate loci bound by MBD6. The authors put forward a model where MBD5/6 protect methylated nuclei for some cell states in a branched developmental trajectory, but allow low-level demethylation of pericentromeric TEs via MBD7 in the VN, speculating that this helps to maintain silencing of some loci in other nuclei via small RNA movement.

Comments:

The study described by this manuscript presents a massive amount of data focused on a specific question related to pollen maturation. I found the manuscript to be both well-written, the experiments thoughtfully designed, and the data to be presented in a relatively accessible way. I do not have major concerns regarding this study, and think it will make a good addition to the Nature Communications journal. I have some minor comments, however:

1. Please don't forget to include raw imaging data to the Dryad database (or other databases).

2. Please include a table that associates barcode information from your snmCT-seq dataset with cell type annotations and other inferred per-cell metrics computed. Alternatively, authors can upload their Seurat (.rds) or Scanpy objects (.h5ad) that include this metadata in their GEO repository.
3. Please check figure legends. The legend for Figure 4 has erroneous labels in the caption.
4. The description of the clusters “MN to VN” and “VN1 to 2” was annotated as such based on the “closest clusters”. Does this mean close in UMAP space? Or do they refer to the markers they show? Additional detail about how these were annotated would be good to include (using a metric other than UMAP coordinates).
5. The authors state that “Expression of MBD7 and DEM was not affected in mbd5/6”, referring to Extended Data Fig. 4e. However looking at this figure, I do see an increase in MBD7 expression (at least in terms of percent of cells expressing) in the VN3 cluster, compared to WT.
6. Authors state that “An unbiased clustering of the snmCT-seq methylation data alone was only able to distinguish VN nuclei from the other nuclei types...” based on cluster information overlaid on a UMAP from methylation data. While I agree that the 2D clustering in UMAP space isn't that strong, there does appear to be differences between some of the cell types. I also would note that this doesn't really constitute “clustering” – perhaps the authors could cluster their data using Louvain or another algorithm and demonstrate this more clearly? I don't think this is important to the overall manuscript, regardless.
7. The authors describe demethylation profiles through VN development using temporally descriptive language (i.e. “immediately after pollen mitosis I” or “decreased rapidly by VN3”), but I don't see how they can relate their pseudotemporal imputation with physical time through development. Do the authors have clear temporal landmarks that they can use to calibrate/align their single-cell data? I would suggest that instead of describing the timing of change in pollen maturation, the authors are describing an order of events, which can have a non-linear relationship to actual pollen age. I know this is nit-picky semantics, but it caught my attention.

Version 1:

Reviewer comments:

Reviewer #1

(Remarks to the Author)

The authors have carefully addressed my concerns. I have no further questions.

Reviewer #2

(Remarks to the Author)

Overall, the authors have comprehensively addressed all my concerns. The new data and analyses, particularly the repeated ChIP experiments and the additional snmCT-seq data from the triple mutant, have significantly enhanced the manuscript's quality and robustness. The revised figures and text present a clearer, more convincing narrative. I fully support the publication of this paper.

The findings of this study provide a thorough understanding of how three MBD proteins actively contribute to epigenetic reprogramming during pollen maturation. The antagonistic interaction between MBD5/6 and MBD7 offers novel insights into the silencing mechanism in the vegetative nucleus (VN). The genetic and multi-omics data presented provide a valuable model and comprehensive resource for the field.

I have one minor suggestion:

Given that MBD5 and MBD6 are primarily expressed in immature VN (Extended Data Fig. 4d), the analysis of the methylation state at the ‘rescued’ or ‘not rescued’ loci (presented in Fig. 5b-c and Extended Data Fig. 9d) would be more appropriately performed using the immature VN single-cell RNA-seq data from stages VN1 to VN3.

Reviewer #3

(Remarks to the Author)

I would like to thank the authors for their thoughtful, and comprehensive replies to mine and (I believe) other reviewers' comments. I also think their updated conclusions given their analyses of additional transgenic lines seem sound. I have no further comment on the manuscript.

Reviewer #1 (Remarks to the Author):

In this work, the authors studied the opposing roles of Arabidopsis MBD5/MBD6 and MBD7 in regulating DNA methylation and transcription. They found that loss of MBD7 rescues transcriptional defects at a subset of loci bound by MBD5/6. Single-nucleus profiling revealed that MBD5/6-bound loci undergo active demethylation in early VN, which is accelerated in *mbd5/6* mutants before transcriptional derepression occurs. In the absence of MBD5/6, MBD7 binds ectopically, correlating with further demethylation and transcriptional activation, which is reversed when MBD7 is lost. Artificial recruitment of MBD7 to MBD5/6 targets similarly induces partial demethylation and transcriptional activation. This study proposes that MBD5/6 maintain silencing by preventing MBD7-mediated enhancement of active demethylation during VN development.

MBD proteins are critical regulators of DNA methylation and transcription, with MBD5/MBD6 and MBD7 known to bind highly methylated genomic regions. However, their functional interactions remain poorly understood. This study presents a compelling model of antagonistic binding between MBD5/MBD6 and MBD7, predominantly in heterochromatic regions, to modulate local DNA methylation and transcription. This research was elegantly designed, featuring well-performed experiments with appropriate controls, thorough data analysis, and conclusions strongly supported by the results. While there are no major concerns, an intriguing question arises regarding H1 expression dynamics during pollen development. It would be insightful to determine the exact timing of H1 absence in early VN, coinciding with DNA methylation changes and the activity of MBD5/6/7, according to their snmRNA-seq data.

We thank Reviewer #1 for their positive evaluation of our manuscript. We agree that the relationship between H1 and MBD5/6/7 dynamics is an interesting question, especially in light of our previous result suggesting that loss of H1 amplifies the *mbd5/6* phenotype in vegetative tissues (Ichino et al. 2022 Cell Reports). Arabidopsis has 3 H1 homologs, H1.1, H1.2, and H1.3. Based on GFP imaging, H1.1 and H1.2 have both been reported to be expressed in early microspores, generative cells, and sperm, but are absent from late microspores and vegetative cells (He *et al.* eLife 2019). H1.3 is not expressed in pollen at any stage (He et al.). Both the snmCT-seq data presented in this manuscript and our previously published 2022 single-cell data are largely consistent with these findings (Figure for Reviewers 1, below). H1.1 and H1.2 are both expressed in microspores (MN) and generative nuclei (GN), while H1.3 is not detected in pollen at any stage. None of the H1 homologs was detected in vegetative nuclei (VN) at any stage. Unlike the GFP data, we only detected mild H1 expression in the sperm (SN) (see Extended Data Figure 4f), but it is possible that the GFP signal detected in SN in He et al. 2019 was due to proteins that had been produced in the GN and persisted after mitosis, since sperm cells are generally transcriptionally quiescent. Overall, both our data and published imaging data suggest that H1 expression and protein levels decrease dramatically prior to pollen mitosis I, so that H1 is already not expressed even in the earliest stages of VN development. Thus, H1 appears to be lost, for the most part, prior to the expression of DME and active demethylation, which occurs early in VN development pseudotime, and also prior to most detectable MBD5/6/7 expression, according to our snmCT-seq data (Fig. 3d-e, Ext. Data 4d, reproduced below) and published 10x data (Ichino et al. 2022). Also, H1 levels are not affected in *mbd5/6* according to our data (Figure for Reviewers 1d). This analysis is now included in the discussion and in Extended Data Figure 4f, shown below.

Text added to the results section:

“Linker histone H1 has also been reported to be depleted in the VN, which is thought to help promote DNA demethylation and loss of chromatin condensation.¹¹ Consistent with this, in both our previous snRNA-seq data⁶ and in the snmCT-seq dataset, H1 expression is present in the

MN but largely absent from VN clusters (**Extended Data Fig. 4f**). Expression of the MBDs, H1, DME, and other DNA methylation-related genes examined was not substantially affected in *mbd5/6* (**Extended Data Fig. 4d-f**).

Text added to the discussion:

“Linker histone H1 is also depleted in the VN.¹¹ H1 is involved in chromatin condensation, so its depletion in VN is thought to facilitate access of DME to some heterochromatic TE targets¹¹. Consistent with this, both our single cell transcriptomic data (**Extended Data Fig. 4f**) and published imaging data¹¹ show that H1.1 and H1.2 expression levels decrease dramatically prior to pollen mitosis I and therefore are already depleted before DME and MBD5/6/7 are upregulated and before active demethylation of the VN begins.”

Figure for Reviewers 1: Expression of H1 homologs across pollen nuclei types. (a) Dotplot showing average expression and % nuclei expressing the three H1 homologs, alongside MBD5, MBD6 and MBD7 for reference. Data shown are all data from wild-type (Col) in the 10x dataset published in Ichino *et al.* 2022. (b) Same as (a), but using our snmCT-seq data from Col instead (this study). (c) UMAP of all Col 10x data from Ichino *et al.* 2022, showing cluster assignments (left), expression of H1.1 (middle), and expression of H1.2 (right). (d) Same as a,b but showing expression of indicated genes in *mbd5/6* nuclei instead of Col nuclei.

Fig. 3d-e: D) Diagram of approximate relative DNA methylation levels during development from vegetative tissue to MN, and finally to mature SN and VN. Grey box shows dynamics revealed in this study. Published data from FACS-sorted populations of MN and mature VN and SN were used for other parts of the diagram^{7,12,13}, and are also consistent with our data. E) Distribution of average per-nucleus smnCT-seq methylation for nuclei assigned to each cluster in (B). Average methylation was computed over regions hypomethylated in the CG context in VN (VN CG hypo DMRs, left) and hypermethylated in the CHH context in VN (VN CHH hyper DMRs, right). These regions were previously identified from bisulfite-sequencing of purified FACS-sorted VN populations¹³. The red horizontal line marks the median methylation level in MN, for reference.

Additional notes to the reviewer:

We would like to note that we have changed our conclusion about MBD7 relocating in response to loss of MBD5/6, based on new data. To validate our initial findings, we repeated the MBD6 and MBD7 ChIP-seq experiment while the paper was under review, with several new independent transgenic lines. We obtained several new lines for each MBD7-Flag background, and selected a set of three to four lines per background with consistent, intermediate expression levels. This includes both new lines and lines used in the previous ChIP experiment. In the new ChIP-seq experiment, the IP efficiency was improved relative to the previous experiment, with strong bands post-IP for all lines (see **Extended Data Fig. 2a**, reproduced below) as well as improved signal/noise ratio in the data. We also censored three lines with unusually strong or weak transgene expression or pulldown efficiency during the ChIP (**Extended Data Fig. 2a**, red stars), leaving two to three lines per genotype. It now seems likely that our previous data, and particularly the MBD7-Flag ChIP-seq samples, had poor capture rates, leading to poor signal/noise ratios. This effect was unfortunately confounded with genotype in the original dataset, where the MBD7-Flag samples in *mbd7* performed worse than in *mbd5/6/7*, leading to our original conclusion of MBD7 relocating in the absence of MBD5/6. These issues have been substantially improved in the new dataset (see **Figure for Reviewers 2**, below, comparing tracks of the old and new

ChIPs), including for transgenic lines that were assayed in both experiments. Therefore, we believe that our previous ChIP was misleading, and we have moved forward with the new data for this study (see Fig. 2, reproduced below).

Extended Data Fig. 2a: Western blot of MBD6-Flag and MBD7-Flag, MBD7-Flag *mbd5/6*, and MBD7-Flag *sln* lines used for ChIP-seq. Left is the blot of chromatin input to the ChIP, right is post-IP. Expected sizes are 35 kDa for MBD6-Flag and 45 kDa for MBD7-Flag, although we always detect bands above these sizes for both proteins. Red stars indicate samples that were censored due to either unusually weak or strong signal in the input relative to the other lines. Each sample was a different independent transgenic line, numbered T2-[#].

Figure for Reviewers 2: Comparison of old and new ChIP-seq data. Tracks are, in order from top: 4 old, followed by 5 new tracks for MBD7-Flag in *mbd7* (blue) and MBD7-Flag *mbd5/6/7* (orange), a track labeling MBD7 peak locations from the new experiment, DNA methylation data from buds (Harris et al. 2018), and gene annotations. Independent

transgenic lines are labeled as T2-[#]. Note that MBD7-Flag line T2-2 and MBD7-Flag *mbd5/6* line T2-4 were run in both the old and new ChIP-seq experiment.

Our new ChIP-seq data is highly consistent between all independent lines. This dataset no longer supports the previous conclusion that MBD7 localization changes in *mbd5/6* (new **Extended Data Fig. 3**, reproduced below). However, we now detect a consistent anticorrelation between MBD5/6 and MBD7 occupancy, which we now explore in depth in the revised manuscript (see revised **Fig. 2**, reproduced below). Ultimately, we conclude that while MBD5/6 doesn't directly impair MBD7 binding, MBD5/6 can bind methylated DNA in deep heterochromatin while MBD7 generally can't. Conversely, MBD6 doesn't bind as strongly in euchromatin as MBD7. However, the subset of *mbd5/6* upregulated transcripts that can be rescued by *mbd7* (Fig. 1d, 'rescued') are strongly co-bound by both MBD5/6 and MBD7 (Fig. 2g). These loci are likely true direct targets of both MBD5/6 and MBD7 in wild-type, but MBD7 appears to only be able to promote their demethylation and overexpression when MBD5/6 are absent. Why exactly that is the case is still unclear, and is the subject of ongoing work in our lab. We apologize for the confusion, and hope that these revised findings are nonetheless convincing.

Figure 2: MBD5/6 and MBD7 prefer different subsets of CG-methylated loci. A) (Left) Heatmaps of methylation density (sum of % methylation across all cytosines in indicated sequence context) in 400 bp windows centered on the midpoint of the peaks shown at right. Each column was normalized separately. (Right) Heatmaps of MBD6-Flag signal and MBD7-Flag signal (\log_2 of IP over no-FLAG control) over the union of all MBD6 and MBD7 peaks. Each column is an independent transgenic line in the indicated mutant background. All heatmaps in (A) share same row order, which

was sorted based on the MBD7-Flag T2-2 sample (third column). The top 10% (MBD7-dominant), bottom 10% (MBD6-dominant), and middle 10% (mixed) of peaks are indicated on the right. B) Example browser tracks showing the location of MBD6/7 union peaks (top), with MBD7-dominant, MBD6-dominant, and mixed peaks labeled, as well as MBD6 and MBD7 ChIP-seq signal (this study; log of IP over no-FLAG control), % DNA methylation data from inflorescences with unopened flower buds in the CG, CHG and CHH context²⁵, gene annotations (black = forward strand, yellow = reverse), transposon (TE) annotations, and peak coordinates. Genes and TEs are from the araport11 annotation.⁴¹ For % DNA methylation, both Col replicates from ²⁵ were pooled, and only sites with 5 or more coverage are shown; small negative value (-10%) corresponds to sites that had coverage, but no methylation, to distinguish from missing data/sites lacking cytosines. C) (left) Representative example of root nucleus expressing pMBD7::MBD7-YFP and incubated with DAPI to stain chromocenters. (right) Percent of foci assayed (n=120 for each condition) where YFP and RFP signal overlapped vs. did not overlap. Left bar shows co-expression of MBD6-RFP and MBD7-YFP, right bar shows control expressing MBD6-RFP with MBD6-YFP. Also see **Extended Data 2d**. D) Percent of MBD7-dominant, mixed, and MBD6-dominant peaks from (A) overlapping annotated transcripts classified as either gene body methylated (gbM, mCG only) or TE-like methylated (teM, mC in all contexts). E) Metaplots of various epigenetic marks (DNA methylation²⁵, NRPE1⁴², histone H1 and H3K9me2⁴³, histone H3 and H3K4me3⁴⁴, accessibility by ATAC-seq⁴⁵), as well as RNA-seq⁴, over the MBD7-dominant, MBD6-dominant, and mixed peaks labeled in A. F) Smoothed distribution of the location of MBD7-dominant, MBD6-dominant, and mixed peaks across Chromosome 4. Approximate location of the centromere shown as a grey bar. G) Metaplots of MBD6 and MBD7 ChIP-seq signal (log₂ of IP over no-FLAG control) over pollen *mbd5/6* upregulated transcripts that were rescued, partially rescued, and not rescued by loss of MBD7 (see clustering in **Fig. 1d**). Control transcripts are an equal number of random genes matched for expression level and length. Plotted values represent the average of multiple independent transgenic lines (N=2 for MBD6-Flag, N=3 for MBD7-Flag).

Extended Data 3: MBD7 ChIP-seq analysis related to Fig. 2. A) Metaplots of various epigenetic marks (histone H2A, H2A.X, and H2A.Z⁴³, H2A.W⁶⁶, H3K27me3⁴⁴, H3K27ac⁶⁷) over the MBD7-dominant, MBD6-dominant, and mixed peaks from Fig. 2a. B) Metaplots and corresponding heatmaps of MBD7-Flag signal (\log_2 of IP over no-FLAG control) in indicated background over transcripts downregulated in *mbd7*, and a set of control genes matched for expression and length. Plotted values represent the average of multiple independent transgenic lines (N=3 for *mbd7*, N=2 for *mbd5/6/7*, and N=3 for *mbd7;sln*). C) Heatmaps of MBD7-Flag signal (\log_2 of IP over no-FLAG control), for multiple independent transgenic lines in the indicated mutant background. Heatmaps all share same row order, which

was sorted based on the MBD7-Flag *mbd7* T2-2 sample (first column), and is the same as in **Fig. 2a**. D) Metaplots and corresponding heatmaps of MBD7-Flag signal (\log_2 of IP over no-FLAG control) in indicated background over pollen *mbd5/6* upregulated transcripts that were rescued, partially rescued, and not rescued by loss of MBD7 (see clustering in **Fig. 1d**). Control transcripts are an equal number of randomly selected genes matched for expression level and length. Plotted values represent the average of multiple independent transgenic lines (N=3 for *mbd7*, N=2 for *mbd5/6/7*, and N=3 for *mbd7;sln*). E) Metaplot and heatmap of MBD6-Flag signal (\log_2 of IP over no-FLAG control) over *mbd5/6* rescued and not rescued transcripts, as well as *mbd7* downregulated transcripts, and their matched control genes. Color legend for metaplot same as in (D).

Extended Data 4: snmCT-seq transcriptome data validation related to Fig. 3. (A) UMAP plots of Col and mbd5/6 samples from both experiments (1 and 2), shown separately. Clusters are labeled on far right plot. Col_1 = Col from expt 1. **(B)** Expression of key marker genes in each cluster, across each experiment and genotype. DML3 and AT5G17340 are elevated in MN. The three VN markers are for early post-mitotic (MSP2), mid-stage (VEX1) and mature (VCK) VN. ENODL7, DUO1 and MORCS mark GN, HTR10/MGH3 and PCR11 mark SN, and GRP17 marks somatic nuclei. **(C)** Heatmap of correlation of RNA-seq data between clusters identified in snmCT-seq (this study) and a prior

10x pollen dataset.⁶ D) Dotplots of average expression of indicated genes across snmCT-seq clusters in Col. E) Dotplots of average expression of indicated genes across snmCT-seq clusters in *mbd5/6*. F) Dotplots of average expression of the three H1 homologs in both published 10x pollen data⁶ and the snmCT-seq data (this study).

Reviewer #2 (Remarks to the Author):

It is well established that during pollen development, the vegetative nucleus (VN) undergoes significant nuclear condensation and epigenetic reprogramming. Previous studies have identified various mechanisms involved in regulating these processes, but their complexity warrants further in-depth investigation. Methyl-CpG-binding domain (MBD) proteins are known to recognize DNA methylation sites and potentially influence subsequent transcriptional activity. Among these, MBD5 and MBD6, which are mildly expressed in the VN, have been implicated in VN function, although their exact mechanisms remain unclear.

Using simultaneous single-nucleus methylcytosine and transcriptome analysis, Picard et al. demonstrated that MBD5 and MBD6 actively repress transcription through DNA methylation in the early VN following the first pollen mitosis. In contrast, their nearly homologous protein, MBD7, counterbalances this repression by promoting DNA demethylation, thereby preventing excessive transcriptional silencing. Notably, the absence of MBD7 can rescue the activation of transposable elements (TEs) in *mbd5/6* mutants. MBD5 and MBD6 maintain transposon silencing in the VN by inhibiting the MBD7 complex from amplifying active demethylation during VN maturation. These findings further support the concept that the VN avoids excessive de-repression through finely tuned regulatory mechanisms, such as the MBD5/6-mediated silencing pathway.

Major concerns:

1. Although exploring how epigenetic reprogramming is regulated during male gametophyte development is an intriguing perspective, the biological significance of MBD5/7-mediated silencing in the VN remains unclear.

We thank the reviewer for raising this point, and apologize for not making this clearer in the manuscript. While much remains to be explored about MBD5/6 and MBD7 specifically, repression of TEs is an evolutionarily conserved biological process required to prevent genomic instability. Dramatic disruptions in genomic stability tend to have strong phenotypes, but mild disruptions can take generations to accumulate to the point of affecting an organism (for example, *met1* and *ddm1* mutants, which take a few generations to develop homeotic defects that cause sterility). These are important defects, but they are not always readily visible under lab conditions. Numerous pathways have evolved to maintain silencing that are often redundant, like MBD5/6 and H1 (Ichino et al. 2022), highlighting the importance of maintaining silencing over these elements. However, as a result of these redundancies, mutations in a single pathway are often asymptomatic.

In our study, we have chosen to explore the function of MBD5/6 and MBD7 during the unusual epigenetic reprogramming that occurs in the VN. This is for two main reasons. First, in the VN, pathways redundant to MBD5/6, like H1, are lost in order to facilitate widespread DNA demethylation (He et al. eLife 2019). Loss of these redundant pathways is presumably why we are able to see a phenotype for *mbd5/6* mutants in the VN, which is not possible in other tissues. Thus, the interaction between MBD5/6 and MBD7 happens to be much easier to study in this unusual tissue. However, we want to point out that MBD5/6 and MBD7 are expressed throughout the plant (Ichino et al. 2022 Cell Rep.), so their function is likely not restricted to the VN, although it may differ in other tissues. Indeed, we previously showed that genetic depletion of H1, which causes chromatin decondensation in vegetative tissues resembling what is seen in wild-type VN,

reveals that MBD5/6 are repressing TEs in seedlings as well, but their function is normally hidden by the redundancy with H1 (Ichino et al. Cell Reports 2022).

The second reason we focused on the roles of MBD5/6/7 during VN reprogramming is that our results suggest that MBD5/6 and MBD7 do in fact play a key role during this process: during the wave of programmed demethylation that occurs during VN maturation, MBD7 helps facilitate demethylation of DME targets, while MBD5/6 help protect loci like transposons from demethylation, ensuring that the correct sites become demethylated while transposons remain (mostly) silent. Our new snmCT-seq data, presented in the updated Fig. 5 (reproduced below) supports this hypothesis. At sites bound by both MBD5/6 and MBD7, it seems that MBD5/6 carries out this function by inhibiting MBD7 function, though the details of how remain unclear. One possibility is that MBD5/6 prevent MBD7 from recruiting its larger complex by oligomerizing (Boone et al. 2023 *Sci Adv*) and physically preventing LIL/IDM3 etc. from accessing the region or creating a permissive chromatin environment for DME. However, we note that relatively few sites are affected by loss of MBD5/6 or MBD7, relative to the number of DME targets, suggesting that even in the VN a lot of redundancy remains. It's likely that other pathways carry out similar functions that remain to be discovered. Another piece of evidence in support of this is the partial rescue of *mbd5/6* by *mbd7*, which suggests that at least one other pathway that promotes DME activity in VN is antagonized by MBD5/6. Thus, while MBD5/6 and MBD7 alone are not essential for VN development, our results suggest that they play a role in an important process, likely alongside other not yet identified pathways that together influence the steady-state outcome of VN epigenetic reprogramming. Disrupting enough of these pathways at the same time would likely have a much stronger effect on pollen development, which is a direction we're eager to explore further. We have added some text to the discussion, emphasizing some of the points above about the potential function of MBD5/6 and MBD7 in pollen development.

Text added to the discussion:

"However, these factors alone are not essential for pollen development, as we did not observe any strong phenotype in any of the mutants. Since MBD5, MBD6 and MBD7 are expressed throughout the plant (Ichino et al. 2022 Cell Rep), we favor the hypothesis that MBD5/6/7 function in most plant cells. However, this does not preclude an important function in pollen. A number of redundant regulatory mechanisms have evolved to reinforce silencing of transposons and other dangerous elements, including MBD5/6, highlighting their importance."

Fig. 5: Loss of MBD7 rescues excessive demethylation in *mbd5/6* pollen. (A) Average CG methylation across the four genotypes tested in smnCT-seq experiment 3, over VN CG hypo DMRs.¹³ Each point in the violin plot represents one nucleus' average methylation over the indicated regions. A red dashed line runs through the wild-type median value. (B) Same as (A), but showing methylation averages across the TSS region of *mbd5/6* upregulated transcripts 'not rescued' (bottom) or 'rescued' (top) by *mbd7*. (A-B) Number of stars indicates Cohen's d. (n.e. = no/minimal effect ($|d| < 0.2$), * = $|d| > 0.2$, ** = $|d| > 0.5$, *** = $|d| > 0.9$, **** = $|d| > 1.5$), red color = $p < 0.001$, two-sided *t*-test. (C) Metaplots of methylation data pseudobulked across all nuclei in each indicated cluster + genotype, over *mbd5/6* upregulated transcripts 'not rescued' (left) or 'rescued' (right) by *mbd7*. (D) Example browser images of *mbd5/6* upregulated

transcripts 'not rescued' (right) or 'rescued' (left) by *mbd7*, showing RNA-seq from whole pollen (Fig. 1), snmCT-seq CG DNA methylation data pooled by indicated genotype + cluster (only MN and VN2 clusters shown; these were selected due to the larger numbers of nuclei in these clusters, which increased the probability of getting good coverage genome-wide after pooling), and ChIP-seq for MBD6 and MBD7-Flag (average of all lines). As much as possible, these loci were chosen based on having fairly even and consistent coverage in all snmCT-seq CG DNA methylation tracks.

2. In Figure 1a, of the 202 transcripts upregulated in the *mbd56* mutant pollen, how many correspond to TE loci and how many to gene loci? Do the *mbd56* and *mbd567* mutants exhibit any pollen-related phenotypic abnormalities or alterations in VN nuclear condensation? Considering the subsequent text mentions that *mbd56*-associated methylation and transcriptional dysregulation primarily occur in the VN, do the *mbd56* and *mbd567* mutants display any abnormalities in pollen tube germination?

We thank the reviewer for raising this point. We explored whether the 202 *mbd5/6* upregulated transcripts were genes or TEs. Our DE analysis used updated annotations based on deep pollen RNA-seq (see Ichino et al. 2022 Cell Reports), and many of the upregulated transcripts we detected represent novel annotations that were not yet classified as genes or TEs. We first assigned each transcript to either 'gene' or 'TE' based on sequence homology using blast: briefly, transcripts with strong homology to a TE along at least 1/3 their length were considered TEs, with all others classified as putative 'genes'. We used the Araport11 gene, TE, and TE gene annotations as the reference for blast (see methods for details). Six novel transcripts with no significant homology to any existing gene or TE annotation were inspected manually, and all appeared to be likely TEs due to strong DNA methylation and NRPE1 occupancy over their entire length. Additionally, while inspecting the other *mbd5/6* upregulated transcripts, we noted that many of both the novel transcripts and the transcripts annotated as genes nonetheless strongly overlapped an annotated TE, or had TE-like DNA methylation or other epigenetic modifications. Based on clustering of DNA methylation, H3K9me2, and NRPE1 levels over each transcript, we estimate that only 27 of the 202 *mbd5/6* upregulated transcripts are *bona fide* expressed genes, with the remainder either having strong sequence homology to existing TE annotations, or having very high DNA methylation and NRPE1 occupancy consistent with silencing. We have added this analysis as a new supplementary figure, **Supp. Fig. 1** (reproduced below).

We have also added this text to the manuscript, in the first results paragraph:

“Nearly all of the *mbd5/6* upregulated targets were either transposons (TEs) or had TE-like DNA methylation patterns (**Supplementary Figure 1**), consistent with MBD5/6 primarily regulating methylated loci. The remaining 20 transcripts were mostly unmethylated, and potentially expressed, genes (**Supplementary Figure 1**, see methods). This suggests that *MBD5* and *MBD6* primarily function to repress TEs in pollen.”

We have not detected any fertility or other pollen defects in *mbd5/6*, although we have not explored pollen tube growth. However, since the majority of the *mbd5/6* misregulated transcripts are TEs or TE-like (Supp. Fig. 1), we do not expect *mbd5/6* to be disrupting the expression of genes important for VN or pollen tube development. As noted in our response to the previous comment, we hypothesize that MBD5/6 primarily function to maintain silencing over transposons and other silenced loci, which becomes particularly important in the VN during the wave of demethylation and transcriptional activation. While loss of *mbd5/6* has no apparent phenotype apart from mild TE upregulation in VN, we hypothesize that other pathways redundantly perform this crucial function, and that disrupting of several at once may affect pollen development.

Supplementary Fig. 1: Annotation of *mbd5/6* upregulated transcripts. (A) Heatmap of normalized average H3K9me2 signal⁶⁰, NRPE1 signal⁴², and DNA methylation in the CG, CHG and CHH contexts²⁵, over all 202 *mbd5/6* upregulated transcripts. As a control, a set of 20 randomly selected control genes is also included (flagged grey in 'up in *mbd5/6*' column). The 'rescue status' column flags loci based on their rescue status in Fig. 1d ('rescued', 'partially rescued', or 'not rescued'; all others flagged as 'inconsistent'). The 'seq. homology' column indicates whether the locus was considered to have higher homology to existing Araport11⁴¹ gene or TE/TE gene annotations (see methods).

Dendrogram and transcript IDs highlighted red are putative expressed genes, based on the hierarchical clustering. (B) Example novel annotation that was considered a gene or gene-like ('novel_Ch2_noncoding_23'). (C) Example novel annotation that was considered a TE, with TE-like methylation and H3K9me2 ('novel_Ch5_noncoding_631').

3. In Fig. 1d and Fig. 2a-d, the authors concluded that MBD7 redistributes to MBD5/6 binding loci, particularly localizing to the pericentromeric regions with higher H3K9me2 levels. Considering that H3K9me2 persists in the early VN before its subsequent depletion, as demonstrated by a recent study (Li et al., 2004; 10.1038/s41467-024-51513-4), were there any changes in H3K9me2 levels in *mbd7*, *mbd5/6*, and *mbd5/6/7* mutants in the early post-mitotic VN?

We understand that the reviewer is suggesting that changes in H3K9me2 levels in *mbd5/6* could be responsible or associated with the redistribution of MBD7 binding to heterochromatin. While this is an interesting point, we no longer believe that MBD7 localization is redistributed in response to loss of MBD5/6. To validate our initial findings, we repeated the MBD6 and MBD7 ChIP-seq experiment while the paper was under review, with several new independent transgenic lines. We obtained several new lines for each MBD7-Flag background, and selected a set of three to four lines per background with consistent, intermediate expression levels. This includes both new lines and lines used in the previous ChIP experiment. In the new ChIP-seq experiment, the IP efficiency was improved relative to the previous experiment, with strong bands post-IP for all lines (see **Extended Data Fig. 2a**, reproduced below) as well as improved signal/noise ratio in the data. We also censored three lines with unusually strong or weak transgene expression or pulldown efficiency during the ChIP (**Extended Data Fig. 2a**, red stars), leaving two to three lines per genotype. It seems likely that our previous data, and particularly the MBD7-Flag ChIP-seq samples, had poor capture rates, leading to poor signal/noise ratios. This effect was unfortunately confounded with genotype in the original dataset, where the MBD7-Flag samples in *mbd7* performed worse than in *mbd5/6/7*, leading to our original conclusion of MBD7 relocalizing in the absence of MBD5/6. These issues have been substantially improved in the new dataset (see **Figure for Reviewers 2**, below), including for transgenic lines that were assayed in both experiments. Therefore, we believe that our previous ChIP was misleading, and we have moved forward with the new data for this study (see **Fig. 2**, reproduced below).

Extended Data Fig. 2a: Western blot of MBD6-Flag and MBD7-Flag, MBD7-Flag *mbd5/6*, and MBD7-Flag *sln* lines used for ChIP-seq. Left is the blot of chromatin input to the ChIP, right is post-IP. Expected sizes are 35 kDa for MBD6-Flag and 45 kDa for MBD7-Flag, although we always detect bands above these sizes for both proteins. Red stars

indicate samples that were censored due to either unusually weak or strong signal in the input relative to the other lines. Each sample was a different independent transgenic line, numbered T2-[#].

Figure for Reviewers 2: Comparison of old and new ChIP-seq data. Tracks are, in order from top: 4 old, followed by 5 new tracks for MBD7-Flag in *mbd7* (blue) and MBD7-Flag *mbd5/6/7* (orange), a track labeling MBD7 peak locations from the new experiment, DNA methylation data from buds (Harris et al. 2018), and gene annotations. Independent transgenic lines are labeled as T2-[#]. Note that MBD7-Flag line T2-2 and MBD7-Flag *mbd5/6* line T2-4 were run in both the old and new ChIP-seq experiment.

Our new ChIP-seq data is highly consistent between all independent lines. This dataset no longer supports the previous conclusion that MBD7 localization changes in *mbd5/6* (new Extended Data Fig. 3, reproduced below). However, we now detect a consistent anticorrelation between MBD5/6 and MBD7 occupancy, which we now explore in depth in the revised manuscript (see revised Fig. 2, reproduced below). Ultimately, we conclude that while MBD5/6 doesn't directly impair MBD7 binding, MBD5/6 can bind methylated DNA in deep heterochromatin while MBD7 generally can't. Conversely, MBD6 doesn't bind as strongly in euchromatin as MBD7. However, the subset of *mbd5/6* upregulated transcripts that can be rescued by *mbd7* (Fig. 1d, 'rescued') are strongly co-bound by both MBD5/6 and MBD7 (Fig. 2g). These loci are likely true direct targets of both MBD5/6 and MBD7 in wild-type, but MBD7 appears to only be able to promote their demethylation and overexpression when MBD5/6 are absent. Why exactly that is the case is still unclear, and is the subject of ongoing work in our lab. We apologize for the confusion, and hope that these revised findings are nonetheless convincing.

Figure 2: MBD5/6 and MBD7 prefer different subsets of CG-methylated loci. A) (Left) Heatmaps of methylation density (sum of % methylation across all cytosines in indicated sequence context) in 400 bp windows centered on the midpoint of the peaks shown at right. Each column was normalized separately. (Right) Heatmaps of MBD6-Flag signal and MBD7-Flag signal (\log_2 of IP over no-FLAG control) over the union of all MBD6 and MBD7 peaks. Each column is an independent transgenic line in the indicated mutant background. All heatmaps in (A) share same row order, which

was sorted based on the MBD7-Flag T2-2 sample (third column). The top 10% (MBD7-dominant), bottom 10% (MBD6-dominant), and middle 10% (mixed) of peaks are indicated on the right. B) Example browser tracks showing the location of MBD6/7 union peaks (top), with MBD7-dominant, MBD6-dominant, and mixed peaks labeled, as well as MBD6 and MBD7 ChIP-seq signal (this study; log of IP over no-FLAG control), % DNA methylation data from inflorescences with unopened flower buds in the CG, CHG and CHH context²⁵, gene annotations (black = forward strand, yellow = reverse), transposon (TE) annotations, and peak coordinates. Genes and TEs are from the araport11 annotation.⁴¹ For % DNA methylation, both Col replicates from ²⁵ were pooled, and only sites with 5 or more coverage are shown; small negative value (-10%) corresponds to sites that had coverage, but no methylation, to distinguish from missing data/sites lacking cytosines. C) (left) Representative example of root nucleus expressing pMBD7::MBD7-YFP and incubated with DAPI to stain chromocenters. (right) Percent of foci assayed (n=120 for each condition) where YFP and RFP signal overlapped vs. did not overlap. Left bar shows co-expression of MBD6-RFP and MBD7-YFP, right bar shows control expressing MBD6-RFP with MBD6-YFP. Also see **Extended Data 2d**. D) Percent of MBD7-dominant, mixed, and MBD6-dominant peaks from (A) overlapping annotated transcripts classified as either gene body methylated (gbM, mCG only) or TE-like methylated (teM, mC in all contexts). E) Metaplots of various epigenetic marks (DNA methylation²⁵, NRPE1⁴², histone H1 and H3K9me2⁴³, histone H3 and H3K4me3⁴⁴, accessibility by ATAC-seq⁴⁵), as well as RNA-seq⁴, over the MBD7-dominant, MBD6-dominant, and mixed peaks labeled in A. F) Smoothed distribution of the location of MBD7-dominant, MBD6-dominant, and mixed peaks across Chromosome 4. Approximate location of the centromere shown as a grey bar. G) Metaplots of MBD6 and MBD7 ChIP-seq signal (log₂ of IP over no-FLAG control) over pollen *mbd5/6* upregulated transcripts that were rescued, partially rescued, and not rescued by loss of MBD7 (see clustering in **Fig. 1d**). Control transcripts are an equal number of random genes matched for expression level and length. Plotted values represent the average of multiple independent transgenic lines (N=2 for MBD6-Flag, N=3 for MBD7-Flag).

Extended Data 3: MBD7 ChIP-seq analysis related to Fig. 2. A) Metaplots of various epigenetic marks (histone H2A, H2A.X, and H2A.Z⁴³, H2A.W⁶⁶, H3K27me3⁴⁴, H3K27ac⁶⁷) over the MBD7-dominant, MBD6-dominant, and mixed peaks from Fig. 2a. B) Metaplots and corresponding heatmaps of MBD7-Flag signal (log₂ of IP over no-FLAG control) in indicated background over transcripts downregulated in *mbd7*, and a set of control genes matched for expression and length. Plotted values represent the average of multiple independent transgenic lines (N=3 for *mbd7*, N=2 for *mbd5/6/7*, and N=3 for *mbd7;sln*). C) Heatmaps of MBD7-Flag signal (log₂ of IP over no-FLAG control), for multiple independent transgenic lines in the indicated mutant background. Heatmaps all share same row order, which

was sorted based on the MBD7-Flag *mbd7* T2-2 sample (first column), and is the same as in Fig. 2a. D) Metaplots and corresponding heatmaps of MBD7-Flag signal (\log_2 of IP over no-FLAG control) in indicated background over pollen *mbd5/6* upregulated transcripts that were rescued, partially rescued, and not rescued by loss of MBD7 (see clustering in Fig. 1d). Control transcripts are an equal number of randomly selected genes matched for expression level and length. Plotted values represent the average of multiple independent transgenic lines (N=3 for *mbd7*, N=2 for *mbd5/6/7*, and N=3 for *mbd7;sln*). E) Metaplot and heatmap of MBD6-Flag signal (\log_2 of IP over no-FLAG control) over *mbd5/6* rescued and not rescued transcripts, as well as *mbd7* downregulated transcripts, and their matched control genes. Color legend for metaplot same as in (D).

4. In Figure 2a, the authors present the signal intensity across all peaks bound by MBD6 and MBD7. To further support the antagonistic roles of these proteins and their functions in pericentromeric heterochromatin, the authors should consider including a heatmap or metaplot to visualize the signal intensity and distribution of MBD6 on MBD7 peaks and vice versa. Additionally, categorizing MBD6 and MBD7 peaks into euchromatic regions and pericentromeric heterochromatic regions could help elucidate changes in binding intensity between the two proteins at their reciprocal peaks. Such analyses would provide deeper insights into the distinct and overlapping functions of MBD6 and MBD7.

Thank you for this suggestion. The new in-depth analysis provided above in Figure 2 addresses the distributions of MBD6 and 7, and we hope this satisfies the reviewer's request.

5. In addition to the MBD6 and MBD7 signals in roots presented in Fig. 2 and Extended Data Fig. 3, have the authors examined their signals and colocalization in the VN? Furthermore, have any changes in the signal levels or foci of MBD7 been observed in *mbd5/6* pollen, despite its transcription levels showing no significant variation?

Given the chromocenter decompaction and the high autofluorescence of pollen, we are not optimistic that we would be able to do this imaging. Also, given the new conclusion that MBD7 is not redistributed in *mbd5/6* we think this experiment is also not as critical. We hope the reviewer agrees.

6. In Fig. 3 and Fig. 4, the authors demonstrated that *mbd5/6* mutants lose additional methylation in the early post-mitotic VN prior to gene upregulation. What is the function of these upregulated genes that require silencing in the early VN, especially considering that their methylation is later lost due to DME activity? Furthermore, MBD5/6 targets were preferentially localized to the highly methylated pericentromeric regions enriched with transposons. Could the premature activation of these transposons in the VN negatively impact VN function or GC development?

We thank the reviewer for this suggestion. The new analysis shown in supplemental fig. 1, as well as our answer to point #2 above, indicates that nearly all of the transcripts that become upregulated in *mbd5/6* are transposons, not genes. It was perhaps misleading for us to call these DEGs (genes), so we have changed this to 'differentially expressed transcripts' or similar throughout the manuscript. Most of these loci are normally silent, consistent with normal transposon silencing, and appear to only undergo demethylation in the VN because of the wave of DNA demethylation that normally occurs during VN development. We speculate that in this unusual chromatin environment, the function of MBD5/6 is specifically to help protect transposons from the wave of DME-mediated demethylation. While this protection is not complete (since these transposons do lose some DNA methylation even in wild-type VN, not just in *mbd5/6*, see Fig. 5b), it seems to be sufficient to prevent transcriptional activation of MBD5/6 upregulated loci in wild-type, as shown in Fig. 1d. It is certainly possible that inappropriate activation of these TEs, as in *mbd5/6*, could have a negative impact on VN function or GC development, although we have not examined this in detail in this study.

Accordingly, the following text has been added to the discussion:

The balance of repression by MBD5/6 and antisilencing by MBD7 together could help ensure low, stable TE expression for siRNA production, while preventing massive loss of silencing that could negatively impact the fitness of the VN. However, these factors alone are not essential for pollen development, as we did not observe any strong phenotype in any of the mutants. Since MBD5, MBD6 and MBD7 are expressed throughout the plant,⁶ we favor the hypothesis that MBD5/6/7 function in most plant cells. However, this does not preclude an important function in pollen. A number of redundant regulatory mechanisms have evolved to reinforce silencing of transposons and other dangerous elements, including MBD5/6, highlighting their importance.

7. Many RNA sequencing analyses, localization experiments, and Co-IP assays should be conducted using pollen rather than inflorescences or roots, as the authors themselves noted that MBD56-mediated silencing exhibits strong cell-type specificity.

We agree that the specificity of the MBD5/6 phenotype means that experiments should be conducted in pollen where possible. All RNA-seq, whole-genome bisulfite sequencing, and single-nucleus multiomic sequencing in this study were done using pollen. However, we have not yet been able to successfully perform FLAG ChIP-seq in pollen. To our knowledge, while histone PTM ChIP from pollen and other low-input tissues have been published (Zhang et al. 2022 Plant Biotechnol J, Li et al. 2024 Nat. Comm), ChIP for FLAG or other non-histone DNA binding proteins have not been achieved. We therefore used inflorescences for our MBD6 and MBD7-FLAG ChIP. For the nuclei imaging, we used roots in order to compare MBD7 dynamics to our other published MBD6-RFP data (Boone et al. 2024 Sci Adv).

As mentioned in point 1, while the *mbd5/6* phenotype is only apparent in the VN of pollen, MBD5/6 and MBD7 are expressed throughout the plant, so we speculate that they function similarly in other tissues, but are redundant with other silencing mechanisms such that the loss of *mbd5/6* alone has no effect outside of the VN. Thus, we anticipate that findings in other tissues are not necessarily irrelevant to MBD5/6/7 mechanisms occurring in the VN.

8. In Extended Data 6A, CHG methylation in the SN is shown to be higher than in the VN. However, according to Walker et al. (2018; 10.1038/s41588-017-0008-5), Calarco et al. (2012; 10.1016/j.cell.2012.09.001), and Ibarra et al. (2012; 10.1126/science.1224839), who performed methylation sequencing based on FACS, total CHG methylation in the VN is reported to be higher than in the SN. Could the authors please explain this discrepancy?

We thank the reviewer for pointing out this discrepancy. We have revised Extended data 6A to reflect the cited literature:

We speculated that the discrepancy between our snmCT-seq data and the published WGBS data from sorted VN nuclei in the studies noted by the reviewer might be due to differences in how we looked at genome-wide DNA methylation. To investigate this, we made heatmaps of methylation along the chromosome length for each nuclei cluster (new Ext Data 6a, reproduced below). Our previous analysis reduced the average methylation across the entire genome to a single value

per nucleus, ignoring pericentromere vs. chromosome arms, whereas these new plots are more similar to the circos plots presented in Walker et al. 2018, and Calarco et al., 2012. The new heatmaps show clear differences in behavior between the pericentromere and the chromosome arms: we see a small increase in CHG methylation in the pericentromeres in the VN compared to SN and MN, consistent with the previous studies noted by the reviewer. However, in the chromosome arms, there is slightly less methylation in the VN (see bottom plot below). Although the difference in the chromosome arms is much smaller, the arms are a much larger proportion of the chromosome than the pericentromere. Thus, when averaging the entire genome into a single value, the small difference in the chromosome arms dominates the overall average, which explains why our genome-wide average plots showed mild CHG methylation loss in VN instead of gain, as reported in Walker et al. 2018, and Calarco et al., 2012.

Interestingly, the increase in CHG methylation relative to MN in the pericentromere is observed as early as in the MNtoVN cluster, and remains mostly stable throughout VN development. This is in contrast with the increase in CHH methylation which occurs progressively during VN maturation. On the other hand, CHG methylation decreases very mildly in GN relative to MN, and further decreases more significantly in SN.

The CHH and CG methylation heatmaps show the expected results, and further highlight how pericentromeric CG methylation does not change in the transition between MN and GN, and only increases later, in the transition between GN and SN.

We have replaced the genome-wide violin plots in Extended data 6C with these heatmaps (now **Ext Data 6a**), as we felt that they were more informative in relation to the published literature.

(Top) Reproduction of **Ext. Data 6a**: Heatmaps of average CG, CHG and CHH methylation for indicated clusters of nuclei across the length of Chr4, with bins of 100kb. Approximate location of the pericentromere is highlighted in diagram above each plot. (Bottom) CHG heatmaps from Ext. Data 6a, with two color scales (up to 80% on the left, 50% on the right)

Despite this pericentromeric increase in CHG methylation in the VN, we still note that that CHG methylation progressively decreases at VN hypo CG DMRs, which are the DME targets. This

suggests that the DME could be responsible for this locus specific decrease in CHG methylation, and is consistent with the reported *in vitro* activity of DME on 5mC in any sequence context (Gehring et al., Cell 2006, <https://doi.org/10.1016/j.cell.2005.12.034>). These new insights have been incorporated in the manuscript as follows:

The introduction about the past literature has been revised as follows:

“The methylome of SN resembles MN, with a small increase in CG methylation and decrease in CHG and CHH methylation (**Fig. 3d, Extended Data Fig. 6a**)^{7,12,13}. The mature VN is strongly hypermethylated in the CHH context relative to MN, with minor CHG methylation increases in the pericentromere, while simultaneously undergoing some loss of CG and CHG methylation due to active demethylation by DME or ROS1 (**Fig. 3d, Extended Data Fig. 6a**)^{7,12,13}.”

The following text has also been added to the results section:

“We first generated chromosome-wide methylation profiles of nuclei in different transcriptionally defined clusters to observe global methylation behavior during pollen development (**Extended Data Fig. 6b**). As expected, global CG methylation levels were largely unchanged across clusters, with a small decrease in the VN lineage and increase in SN relative to GN and MN (**Extended Data Fig. 6b**). This is consistent with previous studies which found CG hypermethylation in the SN.^{7,12,33} Our data indicates that this occurs after pollen mitosis II, since it is not apparent in the GN cluster (**Extended Data Fig. 6b**). Pericentromeric CHG methylation was mildly increased in the VN lineage relative to MN, but decreased in SN relative to GN (**Extended Data Fig. 6b**), also consistent with previous reports.^{7,12,33} Interestingly, the CHG methylation increase relative to MN was apparent as early as in the MNtoVN cluster, and remained mostly stable throughout VN development (**Extended Data Fig. 6b**). Pericentromeric CHH methylation strongly increased in the VN lineage as expected from previous studies,^{7,12,33} and this methylation gain appeared to be progressive, occurring gradually along the VN maturation trajectory (**Extended Data Fig. 6b**).

We next examined methylation patterns in our wild-type nuclei over previously identified regions with less CG methylation in VN compared to SN (“CG hypo DMRs”, which are likely DME targets) and more CHH methylation in VN compared to SN (CHH hyper DMRs) based on FACS-sorted populations¹³. As expected^{7,12,13}, CG methylation over CG hypo DMRs was dramatically decreased in the wild-type VN nuclei, whereas CHH methylation over CHH hyper DMRs decreased in MN, remained low in GN and SN, and greatly increased in the VN (**Fig. 3d-e, Extended Data Fig. 6c**).¹³ CHG methylation levels at CG hypo DMRs were also mildly decreased, despite the increase observed at pericentromeres (**Extended Data Fig. 6b**). This is consistent with the reported *in vitro* activity of DME on cytosine methylation in any sequence context.³⁴ DME was previously reported to mainly target TEs in the chromosome arms in the vegetative cell.”

9. The authors should consider including snmCT-seq results from the VN of the *mbd567* mutant and compare the methylation levels in the VN between *mbd56* and *mbd567* mutants, rather than focusing solely on the methylation levels in mature pollen of the *mbd567* mutant.

We thank the reviewer for this suggestion, and we agree that snmCT-seq data from *mbd5/6/7* would strengthen our conclusions. As suggested, we have now repeated our snmCT-seq experiment, this time including 4 genotypes: Col, *mbd7*, *mbd5/6*, and *mbd5/6/7*. The new data is presented in Figure 5, Supplementary Table 3, and Extended Data Fig. 9, and the previous Fig. 5 analysis based on bulk pollen WGBS has been simplified and is now presented in Extended Data Fig. 8. The new Fig. 5 and Ext. Data Fig. 9 are reproduced below.

The Col and *mbd5/6* data from this new replicate was highly consistent with our previous results (Ext Data Fig. 9a-c). Consistent with our model, we found that the demethylation in *mbd5/6* VN was strongly but partially rescued in *mbd5/6/7* VN at CG hypo DMRs, similar to the partial transcriptional rescue we observe in Fig. 1d. At the transcriptionally “rescued” targets (Fig. 1d),

mbd5/6 hypomethylation was fully rescued, but at the “not-rescued” targets the hypomethylation was only partially rescued (Fig. 5b). Moreover, we found that while methylation levels were unchanged in *mbd7* at the MBD5/6 upregulated loci (Fig. 5b), an increase in methylation can be detected in *mbd7* at the CG hypo DMRs (Fig. 5a), which are putative DME targets, and at the loci that are downregulated in *mbd7* RNA-seq (Ext. Data Fig. 9e). This result suggests that MBD7 helps recruits the demethylation machinery in the VN, as has been proposed in other tissues (Lang et al. 2015 *Mol Cell*), and that this function of MBD7 is normally antagonized by MBD5/6.

We also made the interesting observation that the “rescued” targets tended to be in regions with higher overall methylation levels compared to ‘not rescued’ loci (Fig. 5b), and underwent minimal demethylation in wild-type VN, suggesting that in *mbd5/6* background, MBD7 protein could be required to facilitate access of demethylases to highly methylated and inaccessible chromatin, while being dispensable at more “fragile” regions that are more easily demethylated without MBD7’s help. We have added these results to the manuscript in the section “**Loss of MBD7 rescues demethylation of a subset of loci in *mbd5/6* pollen**”. An excerpt is shown below:

“Wild-type and *mbd5/6* methylation patterns across nuclei clusters in our third snmCT-seq experiment were highly consistent with our previous results, with comparable loss of CG methylation and gain of CHH methylation over VN CG hypo and CHH hyper DMRs¹³ in wild-type VN, as well as additional loss of CG methylation at VN CG hypo DMRs in *mbd5/6* VN nuclei (**Extended Data Fig. 9c**). At VN CG hypo DMRs, the additional demethylation in *mbd5/6* VN was strongly, but not fully, rescued in *mbd5/6/7* VN (**Fig. 5a**). Interestingly, the *mbd7* single mutant showed mildly increased CG methylation levels relative to wild-type in the VN, but not in other nuclei types, suggesting that MBD7 facilitates the wave of CG demethylation during VN development (**Fig. 5a**). We next looked at methylation patterns at transcriptionally ‘rescued’ loci compared to ‘not rescued’ loci (**Fig. 1d**). We found that most of the methylation loss in *mbd5/6* VN was strongly rescued in *mbd5/6/7* at transcriptionally ‘rescued’ loci (**Fig. 5b-d, Extended Data Fig. 9d**). These loci also tended to be in regions with higher overall methylation levels compared to ‘not rescued’ loci, and underwent minimal demethylation in wild-type VN that was restricted to the TSS-proximal region (**Fig. 5b-c, Extended Data Fig. 9d**). This suggests that these loci are mostly in silent chromatin that is strongly protected from active demethylation in VN. The ‘not rescued’ loci were generally less methylated than ‘rescued’ loci in wild-type, and lost substantially more methylation in wild-type VN over both the promoter/TSS region and the 5’ end of the annotated transcript (**Fig. 5b-c, Extended Data Fig. 9d**). Hypomethylation of these regions in *mbd5/6* was generally modest, and was rescued in *mbd5/6/7*, but not as fully as at transcriptionally ‘rescued’ loci (**Fig. 5b-c, Extended Data Fig. 9d**).”

Fig. 5: Loss of MBD7 rescues excessive demethylation in *mbd5/6* pollen. (A) Average CG methylation across the four genotypes tested in snmCT-seq experiment 3, over VN CG hypo DMRs.¹³ Each point in the violin plot represents one nucleus' average methylation over the indicated regions. A red dashed line runs through the wild-type median value. (B) Same as (A), but showing methylation averages across the TSS region of *mbd5/6* upregulated transcripts 'not rescued' (bottom) or 'rescued' (top) by *mbd7*. (A-B) Number of stars indicates Cohen's *d*. (n.e. = no/minimal effect ($|d| < 0.2$), * = $|d| > 0.2$, ** = $|d| > 0.5$, *** = $|d| > 0.9$, **** = $|d| > 1.5$), red color = $p < 0.001$, two-sided *t*-test. (C) Metaplots of methylation data pseudobulked across all nuclei

in each indicated cluster + genotype, over *mbd5/6* upregulated transcripts 'not rescued' (left) or 'rescued' (right) by *mbd7*. (D) Example browser images of *mbd5/6* upregulated transcripts 'not rescued' (right) or 'rescued' (left) by *mbd7*, showing RNA-seq from whole pollen (**Fig. 1**), snmCT-seq CG DNA methylation data pooled by indicated genotype + cluster (only MN and VN2 clusters shown; these were selected due to the larger numbers of nuclei in these clusters, which increased the probability of getting good coverage genome-wide after pooling), and ChIP-seq for MBD6 and MBD7-Flag (average of all lines). As much as possible, these loci were chosen based on having fairly even and consistent coverage in all snmCT-seq CG DNA methylation tracks.

Extended Data 9: Validation of snmCT-seq experiment testing rescue of *mbd5/6* by *mbd7*. (A) UMAPs showing data from each of the four samples from this experiment, projected onto the UMAP from previous experiments 1 + 2 (Fig. 3b). (B) Expression of key marker genes in each cluster (based on projection in A), across each of the four samples in this experiment. DML3 and AT5G17340 are elevated in MN. The three VN

markers are for early post-mitotic (MSP2), mid-stage (VEX1) and late-stage (VCK) VN. ENODL7, DUO1 and MORC5 mark GN, HTR10/MGH3 and PCR11 mark SN, and GRP17 marks somatic nuclei. (C) Comparison of methylation dynamics across all snmCT-seq replicates for Col and *mbd5/6*. CG (top) and CHH (bottom) methylation levels over VN CG hypo DMRs and VN CHH hyper DMRs¹³, across nuclei in indicated cluster. Each point in the violin plot represents one nucleus' average methylation over the indicated DMRs. Nuclei from each of the three snmCT-seq experiments (experiments 1 and 2 shown in Fig. 3, experiment 3 in Fig. 5) are shown separately. (D) Average methylation across the four genotypes tested in snmCT-seq experiment 3, across the TSS region of *mbd5/6* upregulated transcripts 'not rescued' (left) or 'rescued' (right) by *mbd7*. Each point in the violin plot represents one nucleus' average methylation over the indicated regions. (E) Same as D, but for loci downregulated in *mbd7* pollen, see Ext. Data Fig. 1e. (D-E) Number of stars indicates Cohen's d. (n.e. = no/minimal effect ($|d| < 0.2$), * = $|d| > 0.2$, ** = $|d| > 0.5$, *** = $|d| > 0.9$, **** = $|d| > 1.5$), red color = $p < 0.001$, two-sided t -test.

10. Are the amino acids in the StkyC domain of MBD6 and MBD7 conserved?

This is an interesting question. The stkyC domains are somewhat conserved, but a lot of the conserved residues are only in similar chemical families, rather than identical. A more in-depth analysis was performed in our previous paper (Boone et al., Science Advances 2023), figure S5b. As shown below, the StkyC domain is conserved, but the MBD7 sequence is more divergent compared to MBD5 and 6. Some of these amino acid differences are likely what confers specificity for interaction with either IDM3 or ACD15. However, we still know little about how the different amino acids in the StkyC domain mediate protein-protein interactions.

11. Was there any phenotypic change or disruption in developmental pathways when the VN exhibited faster depletion of the CG methylome in the *mbd5/6* background?

We did not detect obvious phenotypic abnormalities in *mbd5/6* pollen. Most of the differentially expressed transcripts were TEs, so the extra methylation loss is unlikely to affect transcription of genes related to developmental pathways. However, we speculate that additional redundancy protects the VN from excessive demethylation, and disrupting several of these pathways at once in addition to MBD5/6 could cause more obvious defects.

12. Based on the observation that not only MBD56, but also heterochromatin factors SUVH4/5/6 and their interacting protein ARID1, which are widely expressed in the VN as shown by a recent study (Li et al., 2024; 10.1038/s41467-024-51513-4), the authors may consider discussing the possible biological significance of these findings.

We understand that the reviewer is suggesting a possible interaction between DNA methyl binders MBD5/6 and H3K9me2 methyltransferases SUVH4/5/6, which also bind methylated DNA, in the VN. This is an interesting hypothesis. We note that MBD5/6 strongly prefer methylation in the CG context (Ext. Data Fig. 2c), while SUVH4/5/6 strongly prefer non-CG methylation. However, many MBD5/6 targets are transposons, which are strongly methylated in all sequence contexts and may be targeted by both MBD5/6 and SUVHs. In our prior IP-MS data from MBD5 and MBD6, we do not recover SUVH4/5/6 or ARID1, so we think a direct interaction is unlikely. We looked at where SUVH4, 5 and 6 are expressed in both the published 10x pollen dataset (Ichino et al. 2022) and the new snmCT-seq data (Figure for Reviewers 2, below).

Figure for Reviewers 2: Expression of H1 homologs across pollen nuclei types. Dotplots showing average expression and % nuclei expressing SUVH4, 5 and 6, and ARID1, alongside MBD5, MBD6 and MBD7 for reference. Data shown are all data from wild-type (WT) or *mbd5/6* mutants in either the 10x dataset published in Ichino *et al.* 2022 or the snmCT-seq data from the current study, indicated at the top of each plot.

The three SUVHs are differentially expressed across pollen cell types, which was also noted in Li et al. 2024. Our data consistently show that SUVH4 is barely expressed in VN, and is instead most strongly expressed in MN and GN, although lowly in both cases. By contrast, SUVH5 is strongly expressed in both MN and immature VN but its expression is lost in mature VN (VN4-5), whereas SUVH6 is strongly expressed in mature VN but not in immature VN or in MN. Loss of SUVH5 expression in VN coincides with the timing of the loss of H3K9me2 reported in Li et al., whereas the function of SUVH6 in mature VN is less clear. SUVH5 is also the only SUVH with detectable expression in the sperm, although low, consistent with reports in Li et al. Unexpectedly, we found the strongest ARID1 expression in the SN and early MN, which is somewhat inconsistent with the immunofluorescence data from Fig. 1 of Li et al. 2024, but is consistent with the proposed function of ARID1 to protect sperm cells from H3K9me2 loss during maturation. MBD5/6/7 expression patterns most resemble those of SUVH5, rather than the ARID1-interactor SUVH6 (Li et al. 2024). So the mechanism described in our work is likely independent of the SUVH6-ARID1 complex described in Li et al. We also note no significant change in the expression of any of these genes due to loss of *mbd5/6*. While more work on understanding the different mechanisms that shape the epigenetic landscape of pollen will be interesting, we have not pursued this particular interaction further, since we believe that this is beyond the scope of this manuscript. We hope the

reviewer agrees. However, we now reference Li et al., 2024 (ref #14) in the following sentence in the introduction:

“The VN has a distinctive epigenetic state, characterized by loss of CG methylation due to active demethylation primarily by DEMETER (DME) and to a lesser extent ROS1, loss of repressive histone modification H3K9me2, and chromatin decondensation⁷⁻¹⁴.”

As well as this sentence in the discussion:

“Several other mechanisms involved in the epigenetic reprogramming of pollen vegetative cells have been described.^{11,14} For example, a recent study found that another set of DNA methyl-binding proteins, SUVH4/5/6, interact with ARID1 to slow down the eviction of H3K9me2 from the VN, functioning in some ways similarly to MBD5/6 in preventing excessive DNA demethylation in VN.¹⁴”

13. Two papers on small RNA movement in pollen should be cited: Wu et al., 2021 (doi: 10.1111/nph.16871) and Herridge et al., 2024 (doi: 10.1038/s41594-024-01392-6).

We thank the reviewer for their suggestion. The two papers (refs 39 and 40) have been cited in the following sentence of the discussion:

“One proposed role for TE expression in the VN is to reinforce silencing in the sperm cells via the generation and movement of small RNAs from the VN to the SN³⁷⁻⁴⁰.”

Reviewer #3 (Remarks to the Author):

In this manuscript, the authors describe an antagonistic relationship between three genes that regulate DNA methylation in pollen. They explored an epigenomic dataset generated for the double mutant, *mbd5/6*, a triple mutant, *mbd5/6/7* and wild-type and found that a large number of loci that are de-methylated and up-regulated in the double mutant were “rescued” by the loss of MBD7. They supported their assertion that MBD7 de-methylates genes that are bound (and kept methylated by) MBD5/6 by using ChIP-seq profiling of MBD6 and MBD7 in WT and mutant lines. They went on to profile different pollen nuclei cell states using the recently-developed snmCT-seq which profiles both cytosine methylation and gene expression of single nuclei. Using this data, they piece together a developmental trajectory and found that demethylation is somewhat specific to the Vegetative Nuclei (VN) and progresses according to developmental time. Lastly, the authors swapped the complex recruitment domain of MBD6 and MBD7, showing that the StkyC domain of MBD7 can ectopically demethylate loci bound by MBD6. The authors put forward a model where MBD5/6 protect methylated nuclei for some cell states in a branched developmental trajectory, but allow low-level demethylation of pericentromeric TEs via MBD7 in the VN, speculating that this helps to maintain silencing of some loci in other nuclei via small RNA movement.

Comments:

The study described by this manuscript presents a massive amount of data focused on a specific question related to pollen maturation. I found the manuscript to be both well-written, the experiments thoughtfully designed, and the data to be presented in a relatively accessible way. I do not have major concerns regarding this study, and think it will make a good addition to the Nature Communications journal. I have some minor comments, however:

1. Please don't forget to include raw imaging data to the Dryad database (or other databases).

We apologize for not including the link in the original submission. Images have been deposited to Dryad. Data will be private until manuscript publication, but can be accessed by reviewers here: http://datadryad.org/stash/share/5uba9L45O928v_qGI_TZaityozQO3nAFwXUI0Jl2BZ4

2. Please include a table that associates barcode information from your snmCT-seq dataset with cell type annotations and other inferred per-cell metrics computed. Alternatively, authors can upload their Seurat (.rds) or Scanpy objects (.h5ad) that include this metadata in their GEO repository.

Thank you for pointing this out. The barcode information is now included in the GEO repository for the snmCT-seq data (filename GSE275831_feature_README.txt). Note that there are 384 barcodes corresponding to one per well of the plate, and each plate uses the same set of barcodes.

Per-nucleus metadata and metrics are in Supplementary Table 3, and include the number of reads obtained per well, RNA-seq statistics (genes detected, etc.), WGBS statistics (percent genome coverage), overall QC pass/fail, final cluster ID based on Seurat, and finally chloroplast and (non-plastid) genome-wide average methylation levels. Additionally, as the reviewer suggested, we have uploaded our Seurat object as a .rds file to the github repository for this project.

3. Please check figure legends. The legend for Figure 4 has erroneous labels in the caption.

Thank you for catching this, Fig. 4 legend has now been fixed, and all other figure legends have been carefully checked.

4. The description of the clusters “MN to VN” and “VN1 to 2” was annotated as such based on the “closest clusters”. Does this mean close in UMAP space? Or do they refer to the markers they show? Additional detail about how these were annotated would be good to include (using a metric other than UMAP coordinates).

This is a good point, and we apologize for not being clear. This labeling was primarily based on the expression pattern of the marker gene set we had been using. “MN to VN” had expression of both MN and early VN markers, but to a lesser extent than either MN or VN1 nuclei. Similarly, “VN1 to 2” had less expression of our early VN marker MSP2 and more expression of mid-VN marker VEX1 than VN1, but more expression of MSP2 and less of VEX1 than the VN2 cluster (see Fig. 3c below). We have clarified this in the methods.

Text added to the methods:

“The “MN to VN” cluster was named based on having intermediate expression of both MN and VN1 markers, and the “VN1 to 2” cluster was named based on having intermediate expression of both VN1 and VN2 markers.”

5. The authors state that “Expression of MBD7 and DEM was not affected in *mbd5/6*”, referring to Extended Data Fig. 4e. However looking at this figure, I do see an increase in MBD7 expression (at least in terms of percent of cells expressing) in the VN3 cluster, compared to WT.

This is an interesting point. It’s true that on average, the expression of *MBD7* across the VN clusters changed a bit in *mbd5/6* – it increased in both VN1 and VN3, while decreasing in VN1to2. Since all of these clusters all correspond to immature VN, we considered this a negligible difference, particularly since the snmCT-seq dataset has many fewer nuclei than the 10x dataset from Ichino et al., 2022, making it overall more noisy. However, to verify this, we checked expression of MBD5, MBD6 and MBD7 in Col and *mbd5/6* in each of the three snmCT-seq replicates:

There does indeed seem to be a shift within each replicate towards earlier expression of MBD7 in *mbd5/6* relative to Col, which is very interesting. However, the expression patterns are fairly noisy at this resolution, and since the observed differences are roughly within the amount of variability we see between replicates of the same genotype, overall we are not confident that this is real. Future work to try to replicate this result using another system, like immunostaining, could confirm if this is the case. Since these differences, if real, are very small, we feel that this can be left to future work without impacting the validity of the current manuscript. We hope the reviewer agrees.

6. Authors state that “An unbiased clustering of the snmCT-seq methylation data alone was only able to distinguish VN nuclei from the other nuclei types...” based on cluster information overlaid on a UMAP from methylation data. While I agree that the 2D clustering in UMAP space isn’t that strong, there does appear to be differences between some of the cell types. I also would note that

this doesn't really constitute "clustering" – perhaps the authors could cluster their data using Louvain or another algorithm and demonstrate this more clearly? I don't think this is important to the overall manuscript, regardless.

We agree that in this analysis, the VN nuclei formed a distinct group with a detectable developmental trajectory, which is consistent with our transcriptome-based clustering. However, we still feel that this analysis alone was not sufficient to confidently differentiate nuclei types/states. This is not to say that DNA methylation cannot theoretically be used to distinguish cell types/states in general. The DNA methylation data for a single nucleus is very sparse, so we are forced to bin the genome into large chunks (25 kb bins in this case) in order to have enough data to analyze per bin. For a genome as small and compact as Arabidopsis, this is a very coarse analysis. DME target regions, for example, tend to be much smaller than that, and so it's unsurprising that this analysis fails to pull out the strong changes we see in our other snmCT-seq DNA methylation analysis. Ultimately, we didn't explore further, given that our analysis based on the transcriptome clusters was clear. We have edited the text in the manuscript to remove the word "clustering" which was improperly used:

"A preliminary dimensionality reduction analysis of the snmCT-seq methylation data aggregated across 25 kb bins genome-wide showed clear separation of VN nuclei from the other nuclei types, suggesting that the VN adopts a particularly unique methylation state (**Extended Data Fig. 5a-c**), consistent with prior reports.^{7,12,13} Other nuclei types were generally not well separated, suggesting single-nuclei methylomes alone are insufficient to identify most pollen nuclei types, at least with the level of resolution possible from low-coverage single-nucleus data."

7. The authors describe demethylation profiles through VN development using temporally descriptive language (i.e. "immediately after pollen mitosis I" or "decreased rapidly by VN3"), but I don't see how they can relate their pseudotemporal imputation with physical time through development. Do the authors have clear temporal landmarks that they can use to calibrate/align their single-cell data? I would suggest that instead of describing the timing of change in pollen maturation, the authors are describing an order of events, which can have a non-linear relationship to actual pollen age. I know this is nit-picky semantics, but it caught my attention.

This is a very good point, and we thank the reviewer for raising it. We have made changes throughout the text to clarify this and to follow the reviewer's suggestion of referring to order of events rather than real time.

Example of changed text in the results section:

"Methylation loss, presumably due to DME activity,^{7,12,13} was already apparent in the VN1 cluster, which in pseudotime represents the 'earliest' non-MN population of nuclei along the VN developmental trajectory, and decreased further in both VN1to2 and VN2 nuclei before mostly plateauing (**Fig. 3e, Extended Data Fig. 6b-e**). By definition, transcriptional changes occurred throughout the progression of MN to VN5 nuclei, since that is how the clusters were identified. However, our data shows that DNA methylation loss due to DME activity plateaus by VN3. This suggests that active removal of CG methylation in VN by DME and/or ROS1¹³ primarily occurs immediately after pollen mitosis I, rather than gradually throughout VN maturation."

Additional notes to the reviewer:

We would like to note that we have changed our conclusion about MBD7 relocalizing in response to loss of MBD5/6, based on new data. To validate our initial findings, we repeated the MBD6 and MBD7 ChIP-seq experiment while the paper was under review, with several new independent

transgenic lines. We obtained several new lines for each MBD7-Flag background, and selected a set of three to four lines per background with consistent, intermediate expression levels. This includes both new lines and lines used in the previous ChIP experiment. In the new ChIP-seq experiment, the IP efficiency was improved relative to the previous experiment, with strong bands post-IP for all lines (see **Extended Data Fig. 2a**, reproduced below) as well as improved signal/noise ratio in the data. We also censored three lines with unusually strong or weak transgene expression or pulldown efficiency during the ChIP (**Extended Data Fig. 2a**, red stars), leaving two to three lines per genotype. It now seems likely that our previous data, and particularly the MBD7-Flag ChIP-seq samples, had poor capture rates, leading to poor signal/noise ratios. This effect was unfortunately confounded with genotype in the original dataset, where the MBD7-Flag samples in *mbd7* performed worse than in *mbd5/6/7*, leading to our original conclusion of MBD7 relocalizing in the absence of MBD5/6. These issues have been substantially improved in the new dataset (see **Figure for Reviewers 2**, below, comparing tracks of the old and new ChIPs), including for transgenic lines that were assayed in both experiments. Therefore, we believe that our previous ChIP was misleading, and we have moved forward with the new data for this study (see **Fig. 2**, reproduced below).

Extended Data Fig. 2a: Western blot of MBD6-Flag and MBD7-Flag, MBD7-Flag *mbd5/6*, and MBD7-Flag *sln* lines used for ChIP-seq. Left is the blot of chromatin input to the ChIP, right is post-IP. Expected sizes are 35 kDa for MBD6-Flag and 45 kDa for MBD7-Flag, although we always detect bands above these sizes for both proteins. Red stars indicate samples that were censored due to either unusually weak or strong signal in the input relative to the other lines. Each sample was a different independent transgenic line, numbered T2-[#].

Figure for Reviewers 2: Comparison of old and new ChIP-seq data. Tracks are, in order from top: 4 old, followed by 5 new tracks for MBD7-Flag in *mbd7* (blue) and MBD7-Flag *mbd5/6/7* (orange), a track labeling MBD7 peak locations from the new experiment, DNA methylation data from buds (Harris et al. 2018), and gene annotations. Independent transgenic lines are labeled as T2-[#]. Note that MBD7-Flag line T2-2 and MBD7-Flag *mbd5/6* line T2-4 were run in both the old and new ChIP-seq experiment.

Our new ChIP-seq data is highly consistent between all independent lines. This dataset no longer supports the previous conclusion that MBD7 localization changes in *mbd5/6* (new **Extended Data Fig. 3**, reproduced below). However, we now detect a consistent anticorrelation between MBD5/6 and MBD7 occupancy, which we now explore in depth in the revised manuscript (see revised **Fig. 2**, reproduced below). Ultimately, we conclude that while MBD5/6 doesn't directly impair MBD7 binding, MBD5/6 can bind methylated DNA in deep heterochromatin while MBD7 generally can't. Conversely, MBD6 doesn't bind as strongly in euchromatin as MBD7. However, the subset of *mbd5/6* upregulated transcripts that can be rescued by *mbd7* (Fig. 1d, 'rescued') are strongly co-bound by both MBD5/6 and MBD7 (Fig. 2g). These loci are likely true direct targets of both MBD5/6 and MBD7 in wild-type, but MBD7 appears to only be able to promote their demethylation and overexpression when MBD5/6 are absent. Why exactly that is the case is still unclear, and is the subject of ongoing work in our lab. We apologize for the confusion, and hope that these revised findings are nonetheless convincing.

Figure 2: MBD5/6 and MBD7 prefer different subsets of CG-methylated loci. A) (Left) Heatmaps of methylation density (sum of % methylation across all cytosines in indicated sequence context) in 400 bp windows centered on the midpoint of the peaks shown at right. Each column was normalized separately. (Right) Heatmaps of MBD6-Flag signal and MBD7-Flag signal (\log_2 of IP over no-FLAG control) over the union of all MBD6 and MBD7 peaks. Each column is an independent transgenic line in the indicated mutant background. All heatmaps in (A) share same row order, which

was sorted based on the MBD7-Flag T2-2 sample (third column). The top 10% (MBD7-dominant), bottom 10% (MBD6-dominant), and middle 10% (mixed) of peaks are indicated on the right. B) Example browser tracks showing the location of MBD6/7 union peaks (top), with MBD7-dominant, MBD6-dominant, and mixed peaks labeled, as well as MBD6 and MBD7 ChIP-seq signal (this study; log of IP over no-FLAG control), % DNA methylation data from inflorescences with unopened flower buds in the CG, CHG and CHH context²⁵, gene annotations (black = forward strand, yellow = reverse), transposon (TE) annotations, and peak coordinates. Genes and TEs are from the araport11 annotation.⁴¹ For % DNA methylation, both Col replicates from ²⁵ were pooled, and only sites with 5 or more coverage are shown; small negative value (-10%) corresponds to sites that had coverage, but no methylation, to distinguish from missing data/sites lacking cytosines. C) (left) Representative example of root nucleus expressing pMBD7::MBD7-YFP and incubated with DAPI to stain chromocenters. (right) Percent of foci assayed (n=120 for each condition) where YFP and RFP signal overlapped vs. did not overlap. Left bar shows co-expression of MBD6-RFP and MBD7-YFP, right bar shows control expressing MBD6-RFP with MBD6-YFP. Also see **Extended Data 2d**. D) Percent of MBD7-dominant, mixed, and MBD6-dominant peaks from (A) overlapping annotated transcripts classified as either gene body methylated (gbM, mCG only) or TE-like methylated (teM, mC in all contexts). E) Metaplots of various epigenetic marks (DNA methylation²⁵, NRPE1⁴², histone H1 and H3K9me2⁴³, histone H3 and H3K4me3⁴⁴, accessibility by ATAC-seq⁴⁵), as well as RNA-seq⁴, over the MBD7-dominant, MBD6-dominant, and mixed peaks labeled in A. F) Smoothed distribution of the location of MBD7-dominant, MBD6-dominant, and mixed peaks across Chromosome 4. Approximate location of the centromere shown as a grey bar. G) Metaplots of MBD6 and MBD7 ChIP-seq signal (log₂ of IP over no-FLAG control) over pollen *mbd5/6* upregulated transcripts that were rescued, partially rescued, and not rescued by loss of MBD7 (see clustering in **Fig. 1d**). Control transcripts are an equal number of random genes matched for expression level and length. Plotted values represent the average of multiple independent transgenic lines (N=2 for MBD6-Flag, N=3 for MBD7-Flag).

Extended Data 3: MBD7 ChIP-seq analysis related to Fig. 2. A) Metaplots of various epigenetic marks (histone H2A, H2A.X, and H2A.Z⁴³, H2A.W⁶⁶, H3K27me3⁴⁴, H3K27ac⁶⁷) over the MBD7-dominant, MBD6-dominant, and mixed peaks from Fig. 2a. B) Metaplots and corresponding heatmaps of MBD7-Flag signal (\log_2 of IP over no-FLAG control) in indicated background over transcripts downregulated in *mbd7*, and a set of control genes matched for expression and length. Plotted values represent the average of multiple independent transgenic lines (N=3 for *mbd7*, N=2 for *mbd5/6/7*, and N=3 for *mbd7;sln*). C) Heatmaps of MBD7-Flag signal (\log_2 of IP over no-FLAG control), for multiple independent transgenic lines in the indicated mutant background. Heatmaps all share same row order, which

was sorted based on the MBD7-Flag *mbd7* T2-2 sample (first column), and is the same as in **Fig. 2a**. D) Metaplots and corresponding heatmaps of MBD7-Flag signal (\log_2 of IP over no-FLAG control) in indicated background over pollen *mbd5/6* upregulated transcripts that were rescued, partially rescued, and not rescued by loss of MBD7 (see clustering in **Fig. 1d**). Control transcripts are an equal number of randomly selected genes matched for expression level and length. Plotted values represent the average of multiple independent transgenic lines (N=3 for *mbd7*, N=2 for *mbd5/6/7*, and N=3 for *mbd7;sln*). E) Metaplot and heatmap of MBD6-Flag signal (\log_2 of IP over no-FLAG control) over *mbd5/6* rescued and not rescued transcripts, as well as *mbd7* downregulated transcripts, and their matched control genes. Color legend for metaplot same as in (D).

Reviewer #1 (Remarks to the Author):

The authors have carefully addressed my concerns. I have no further questions.

We thank the reviewer for their constructive comments, and for their positive assessment of the revised manuscript.

Reviewer #2 (Remarks to the Author):

Overall, the authors have comprehensively addressed all my concerns. The new data and analyses, particularly the repeated ChIP experiments and the additional snmCT-seq data from the triple mutant, have significantly enhanced the manuscript's quality and robustness. The revised figures and text present a clearer, more convincing narrative. I fully support the publication of this paper.

The findings of this study provide a thorough understanding of how three MBD proteins actively contribute to epigenetic reprogramming during pollen maturation. The antagonistic interaction between MBD5/6 and MBD7 offers novel insights into the silencing mechanism in the vegetative nucleus (VN). The genetic and multi-omics data presented provide a valuable model and comprehensive resource for the field.

I have one minor suggestion:

Given that MBD5 and MBD6 are primarily expressed in immature VN (Extended Data Fig. 4d), the analysis of the methylation state at the 'rescued' or 'not rescued' loci (presented in Fig. 5b-c and Extended Data Fig. 9d) would be more appropriately performed using the immature VN single-cell RNA-seq data from stages VN1 to VN3.

We thank the reviewer for their constructive comments, and for their positive assessment of the revised manuscript.

We investigated the possibility of using only 'immature' VN differentially expressed transcripts for the analyses in Fig. 5b-c, and Ext Data 9 (now supp fig. 10). We agree that this would be a more fair comparison; however, there are relatively few upregulated transcripts in *mbd5/6*, and fewer that were clearly assigned to 'rescued' or 'not rescued' ($n = 100$ and $n = 53$ respectively). Subsetting these transcripts further to only consider the ones also upregulated in immature VN reduces these numbers to $n = 15$ and $n = 17$ (rescued and not rescued, respectively) which is too few to draw conclusions. We have therefore decided to leave Fig. 5b-c and Ext Data 9 (now supp fig 10) unchanged.

Reviewer #3 (Remarks to the Author):

I would like to thank the authors for their thoughtful, and comprehensive replies to mine and (I believe) other reviewers' comments. I also think their updated conclusions given their analyses of additional transgenic lines seem sound. I have no further comment on the manuscript.

We thank the reviewer for their constructive comments, and for their positive assessment of the revised manuscript.